# The equatorial position of the metaphase plate ensures symmetric cell divisions

Chia Huei Tan[1,2], Ivana Gasic[1,2†], Sabina P Huber-Reggi[2†], Damian Dudka[1], Marin Barisic[3], Helder Maiato[3,4], Patrick Meraldi[1]*

[1]Department of Physiology and Metabolism, Medical Faculty, University of Geneva, Geneva, Switzerland; [2]Institute of Biochemistry, ETH Zurich, Zurich, Switzerland; [3]Instituto de Biologia Molecular e Celular, Universidade do Porto, Porto, Portugal; [4]Cell Division Unit, Department of Experimental Biology, Faculdade de Medicina, Universidade do Porto, Porto, Portugal

**Abstract** Chromosome alignment in the middle of the bipolar spindle is a hallmark of metazoan cell divisions. When we offset the metaphase plate position by creating an asymmetric centriole distribution on each pole, we find that metaphase plates relocate to the middle of the spindle before anaphase. The spindle assembly checkpoint enables this centering mechanism by providing cells enough time to correct metaphase plate position. The checkpoint responds to unstable kinetochore–microtubule attachments resulting from an imbalance in microtubule stability between the two half-spindles in cells with an asymmetric centriole distribution. Inactivation of the checkpoint prior to metaphase plate centering leads to asymmetric cell divisions and daughter cells of unequal size; in contrast, if the checkpoint is inactivated after the metaphase plate has centered its position, symmetric cell divisions ensue. This indicates that the equatorial position of the metaphase plate is essential for symmetric cell divisions.

*For correspondence: Patrick. meraldi@unige.ch

†These authors contributed equally to this work

Competing interests: The authors declare that no competing interests exist.

## Introduction

During mitosis, chromosomes are bound to microtubules emanating from both poles of the mitotic spindle via sister-kinetochores and aligned on the metaphase plate precisely in the middle of the spindle. The equatorial position of the metaphase plate is a distinctive feature of metazoan, plant, and many fungal cells. A centered metaphase plate is established even in asymmetrically dividing cells, where daughter cells of unequal size are obtained by an asymmetric positioning of the spindle prior to anaphase (e.g., in *Caenorhabditis elegans* embryos) or an asymmetric elongation of the spindle in anaphase (e.g., in *Drosophila melanogaster* embryonic neuroblasts [*Kaltschmidt et al., 2000*; *Schneider and Bowerman, 2003*]). However, the reason why the metaphase plate is located in the middle of the spindle is not known. One hypothesis is that the centered position facilitates the synchronous arrival of chromosomes at spindle poles during anaphase to prevent chromosomes from being caught on the wrong side of the cytokinetic furrow (*Nicklas and Arana, 1992*; *Goshima and Scholey, 2010*). Elegant work in meiotic praying mantis cells demonstrated that the equatorial positioning of the metaphase plate is not a mere consequence of bipolar kinetochore–microtubule attachments, as trivalent sex-chromosome align in the middle of the spindle, even though trivalent attachment does not favor an equatorial position (*Nicklas and Arana, 1992*). Moreover, previous studies in *Chlamydomonas rheinhardtii* and *C. elegans* showed that an asymmetry in centriole numbers at spindle poles led to an asymmetric metaphase plate position, even though chromosomes established bipolar attachments (*Greenan et al., 2010*; *Keller et al., 2010*). While in algae, longer half-spindles were associated with the pole containing fewer centrioles, in nematodes, longer half-spindles emanated from the pole containing more centrioles. However, whether cells react to

**eLife digest** The genetic information of a cell is stored in the form of chromosomes. Before a cell divides, its entire set of chromosomes is duplicated so that the two newly formed daughter cells receive a full set. In animal cells, the chromosomes line up in the center of the cell. The two sets of chromosomes are then separated by a structure known as the spindle, which attaches long filaments of proteins to the chromosomes and pulls one set to either side of the cell. Birth defects and cancer can result from a cell ending up with too many, or too few, chromosomes. Therefore, a safety mechanism called the spindle assembly checkpoint ensures that all of the chromosomes have correctly attached to the spindle before chromosome separation begins.

Although it has been known for over a hundred years that chromosomes line up precisely in the center of the cell before they are separated, the reason why this occurs has remained unknown. Tan et al. investigated this problem by altering human cells so that the chromosomes did not align in the middle of the cell, but instead lined up off-center. However, after a short delay the chromosomes relocated to the center.

Further investigation revealed that the spindle assembly checkpoint gives cells the time required to re-position the chromosomes in the center of the cell. When the chromosomes are off-center, their connections to the spindle are altered. The spindle assembly checkpoint detects these changes and delays chromosome separation until these errors are corrected. When Tan et al. inactivated the spindle assembly checkpoint before the chromosomes had time to align at the center, the cells divided to produce two unequal cells.

This study shows that the central position of chromosomes is essential for cells to divide into two equal cells. After understanding why this is, the next big challenge will be to find out how a cell re-positions chromosomes that are off-center and how it places them precisely in the middle of the cell.

asymmetrically located metaphase plates and the long-term consequences of this asymmetry are not known. Here, we investigated these questions in human tissue culture cells. We find that cells correct metaphase plate position before anaphase onset, we demonstrate that a centered metaphase plate position relies on the spindle assembly checkpoint (SAC) to provide sufficient time for this correction mechanisms, and we show that a failure to correct plate position leads to asymmetric cell divisions.

## Results

### Cells center the metaphase plate position before anaphase onset

To monitor the relative position of the metaphase plate in the spindle over time, we recorded by time-lapse imaging HeLa cells stably expressing eGFP-centrin1 (centriole marker) and eGFP-CENPA (kinetochore marker) and automatically tracked centrosomes and the metaphase plate using an in-house developed software (*Jaqaman et al., 2010*; *Vladimirou et al., 2013*). Metaphase or late prometaphase cells were recorded over a short period of 5 min in 3D at a resolution of 7.5 s under conditions of low phototoxicity compatible with anaphase entry (*Jaqaman et al., 2010*). By plotting the ratio R of the half-spindle lengths of metaphase cells at the onset of our recordings (first three time points), we found a broad distribution centered around median R = 0.98, which represents nearly equal half-spindle lengths. When analyzing the subset of cells that entered anaphase during our recordings 30 s before anaphase, we found a sharp R distribution in the middle of the spindle (median R = 1.02; *Figure 1A*): less than 10% of the R values were smaller than 0.85 or larger than 1.15 at anaphase onset, while in the metaphase population over 24.2% were outside of these boundaries. This suggested a centering mechanism for the metaphase plate as cells progressed towards anaphase. To test this hypothesis, we aimed to create asymmetric spindles by generating cells with an asymmetric centriole distribution, using small interfering (si)RNAs against Sas-6, a protein required for centriole duplication (*Leidel et al., 2005*). This procedure was used on a set of HeLa eGFP-centrin cells that co-expressed either eGFP-CENPA, α-tubulin-mRFP (spindle marker), or Histone H2B-mRFP (chromosome marker). Every wild-type mitotic cell contains four centrioles: one oldest (grandmother) centriole, one older (mother) centriole, and their two respective daughter centrioles (*Nigg and Raff, 2009*), which all have different eGFP-centrin signal intensities (*Kuo et al., 2011*). A 24-hr Sas-6 depletion led to a mix of cells with two centrioles per pole,

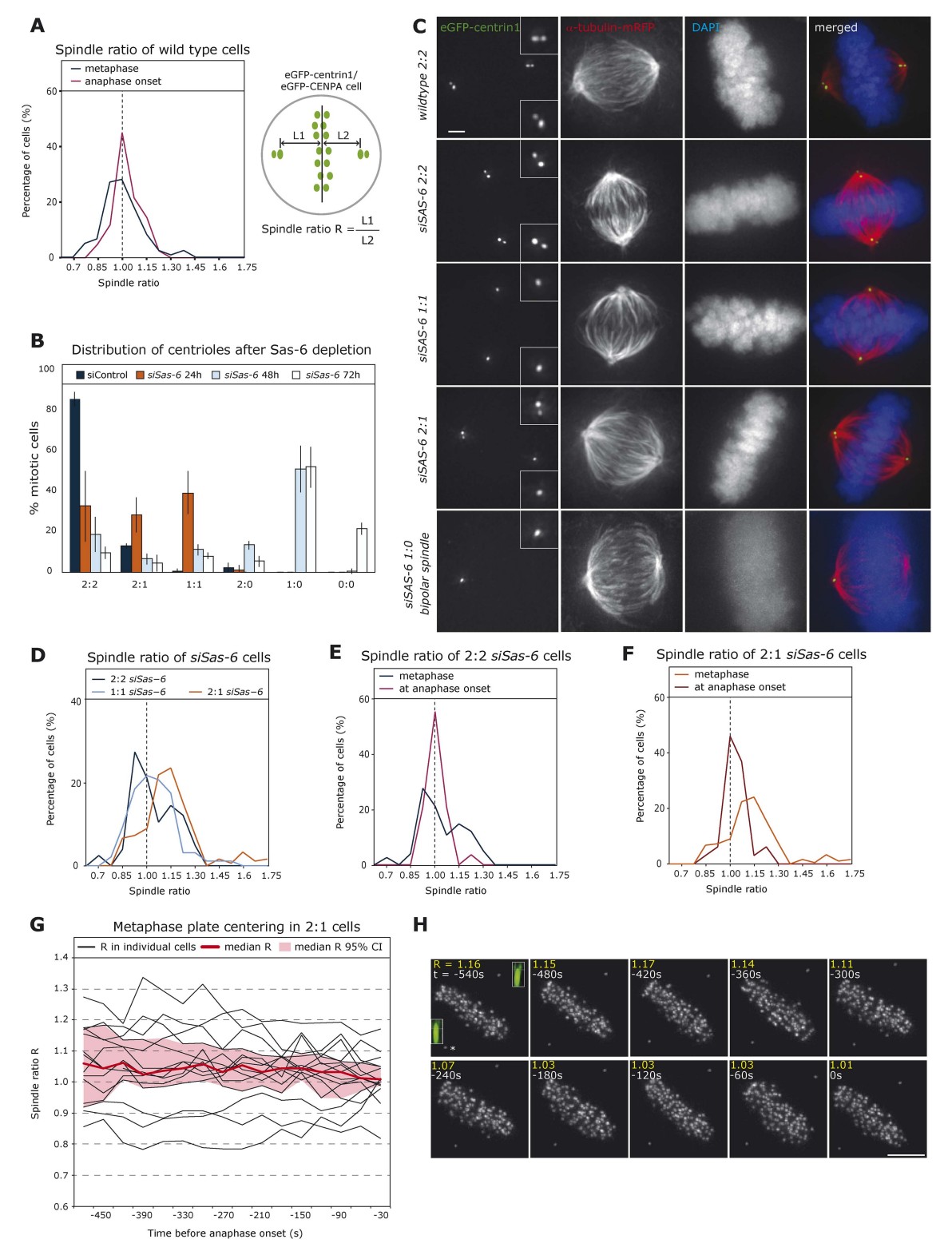

**Figure 1**. Cells center the position of the metaphase plate before anaphase onset. (**A**) Distribution of spindle ratio R in metaphase cells in wild-type eGFP-centrin1/CENPA HeLa cells during metaphase in general (black curve) or 30 s before anaphase onset (red curve). The spindle ratio R was calculated by dividing the half-spindle length L1 associated with the grandmother centriole (brightest eGFP-centrin1) by the other half-spindle length L2 (for all cell numbers in all experiments see **Table 1**). (**B**) Depletion of Sas-6 but not control depletion leads to the gradual loss of centrioles after 24, 48, or 72 hr,
*Figure 1. continued on next page*

*Figure 1. Continued*

resulting in a mixed population of cells with different number of centrioles as indicated. Centrioles were visualized based on images of eGFP-centrin1/mRFP-α-tubulin cells as shown in **C** (n = 50 cells per experiment, in 3 experiments, error bars indicate s.e.m.). (**C**) Immunofluorescence images of eGFP-centrin1 (green)/mRFP-α-tubulin (red) cells stained with DAPI (blue) with different centriole configurations as indicated. Scale bar indicates 2 μm. (**D**) Distribution of spindle ratio R in metaphase cells in *siSas-6*-transfected eGFP-centrin1/CENPA cells with different centriole configurations (2:2, 2:1, or 1:1). The spindle ratio was calculated by dividing the half-spindle length L1 associated with the grandmother centriole (2:2 and 1:1 cells) or the half-spindle length associated with 2 centrioles (2:1 cells) by the other half-spindle length L2. The spindle ratio of 2:1 cells was significantly different from the ratios seen in 2:2 or 1:1 cells (T-test with Welch's correction, 2:1 > 2:2, p = 0.018). (**E, F**) Distribution of spindle ratio R in Sas-6-depleted eGFP-centrin1/CENPA 2:2 (**E**) or 2:1 (**F**) cells in metaphase and before anaphase onset. 2:1 cells have a significantly more asymmetric plate position in metaphase when compared to cells just before anaphase (Mann–Whitney U test, p = $7.16 \times 10^{-7}$). (**G**) Plot of half-spindle ratio R over time in 15 individual eGFP-centrin1/CENPA 2:1 cells that entered anaphase during live-cell imaging (black lines). The timeline was synchronized to anaphase onset (t = 0); the red curve indicates the median of R, and red surface the 95% confidence interval. Note how median R approaches 1 over time and how its variability decreases. (**H**) Time-lapse images of a eGFP-centrin1/CENPA 2:1 cell as analyzed in **G**. Half-spindle ratio R and time before anaphase are indicated for each frame. Number of centrioles was determined in IMARIS in 3D (see 3D-insets in green), * denotes the pole with 1 centriole.

one centriole per pole, or one pole with one centriole and the other pole with two centrioles (called from here on 2:2, 1:1, or 2:1 cells; *Figure 1B,C*). Our intensity measurements revealed that in 2:1 cells it was most often the oldest centriole that gave rise to a daughter centriole, probably due to limiting levels of Sas-6 (data not shown). Tracking of HeLa eGFP-centrin1/CENPA cells indicated that the distribution of half-spindle ratios during metaphase was broad in 2:2 or 1:1 cells, but that on average the plate was located in the middle of the cell (median R = 1.04 (2:2) and 1.03 (1:1); note that 2:2 cells served as control transfection for all subsequent experiments; to be consistent in 2:2 or 1:1 cells, R was calculated as the length of the half-spindle associated to the grandmother centriole divided by the opposite half-spindle length, while in 2:1 cells, R represents the ratio of the half-spindle associated to the 2-centriole pole over the opposite half-spindle length; *Figure 1D*). As in wild-type cells, the range of R had narrowed by the time cells were about to enter anaphase, consistent with a centering process (*Figure 1E*). In 2:1 cells, we found two half-spindles of unequal length, with a shorter half-spindle associated with the pole containing one centriole (median R = 1.12, p < 0.0001 in Mann–Whitney U test compared to symmetric distribution; R > 1.15 in 42.9% of the cells; *Figure 1D*). This asymmetry was corrected before anaphase onset (median R = 1.03; *Figure 1F*), consistent with the existence of a centering process. Since our videos were much shorter than the overall duration of metaphase, we were unable to directly visualize the centering process. To circumvent this difficulty, we recorded longer time series of 2:1 cells, monitoring them for 15 min at 30-s intervals. By plotting R over time in 15 cells that spend at least 7 min in metaphase before entering anaphase, we could directly observe how cells centered the plate position over time, while reducing the variability in metaphase plate position (*Figure 1G,H*).

**Table 1.** Number of cells in every experiment

| Condition | N° of cells |
| --- | --- |
| WT | 40 |
| WT at anaphase onset | 42 |
| WT + MPS1-IN | 19 |
| *SiSas-6* 2:2 cells | 41 |
| *SiSas-6* 2:2 cells at anaphase onset | 29 |
| *SiSas-6* 2:2 cells + MPS1-IN | 19 |
| *SiSas-6* 2:1 cells | 59 |
| *siSas-6* 2:1 cells + MG132 | 18 |
| *SiSas-6* 2:1 cells at anaphase onset | 33 |
| *SiSas-6* 2:1 cells in long term videos | 14 |
| *SiSas-6* 2:1 cells + MPS1-IN | 26 |
| *SiSas-6* 1:1 cells | 36 |
| *SiKif2a + siMCAK* | 20 |
| *SiKif2a + siMCAK* at anaphase onset | 41 |
| *SiKif2a + siMCAK + siSas-6* 2:1 cells | 29 |
| *SiKif2a + siMCAK + siSas-6* 2:1 cells at anaphase onset | 24 |
| *SiSas-6* 2:1 cells + ZM1 | 10 |
| *SiSas-6* 2:2 + DMSO | 15 |
| *SiSas-6* 2:2 + taxol | 11 |
| *SiSas-6* 2:1 + DMSO | 16 |
| *SiSas-6* 2:1 + taxol | 15 |
| Centriole laser-ablation (2:1) | 11 |
| Control laser-ablation | 8 |
| Centriole laser-ablation (2:1) + Mps1 inhibitor | 6 |
| Control laser-ablation + Mps1 inhibitor | 6 |

## The SAC provides cells with sufficient time to center the metaphase plate

To investigate how cells center the metaphase plate, we analyzed the timing of 2:1 cells in comparison to 2:2 or 1:1 cells by monitoring either HeLa eGFP-centrin2/H2B-mRFP or HeLa eGFP-centrin1/mRFP-α-tubulin cells. A 2:2, 1:1, or a 2:1 centriole configuration affected neither the timing of bipolar spindle assembly nor the time it took to align all chromosomes on the metaphase plate (*Figure 2A,B*). Anaphase onset, however, was delayed by 12 min (*Figure 2C*, p = 0.003 in Mann–Whitney U test) in 2:1 cells, when compared to 2:2 or wild-type cells, resulting in 2:1 cells spending twice the amount of time in metaphase (*Figure 2D*, p = 0.015 in Mann–Whitney U test). Anaphase time in 2:1 cells was also delayed in eGFP-centrin1/mRFP-α-tubulin cells 2:1 cells (*Figure 2E*), indicating that it is a robust phenomenon (p = 0.003 in Mann–Whitney U test). In contrast, anaphase was not delayed in 1:1 cells, indicating that the longer metaphase timing seen in 2:1 cells was not caused by the loss of daughter centrioles, but the consequence of centriole asymmetry (*Figure 2C,E*). Anaphase onset is under the control of the SAC, which inhibits the anaphase-promoting complex/cyclosome and delays anaphase onset if kinetochores are not properly attached to spindle microtubules (*Khodjakov and Pines, 2010*; *Foley and Kapoor, 2013*). To test whether the observed metaphase delay depended on the SAC, we co-depleted the SAC protein Mad2 or inhibited the SAC kinase Mps1 in Sas-6-depleted cells (*Figure 2—figure supplement 1*). In the absence of Mad2, 2:2, 2:1, 1:1, or control-treated cells had the same anaphase timing, indicating that the anaphase delay in 2:1 cells depends on the SAC (*Figure 2F*). The same result could be found in Mps1-inhibited cells (*Figure 2—figure supplement 2*). To test whether metaphase plate position is corrected as a result of the additional time provided by the SAC, we prolonged metaphase in *siSas-6*-treated cells by 1 hr by adding the proteasome inhibitor MG132 and found a symmetric plate position in 2:1 cells (median R = 0.98; *Figure 2G*). In contrast, when we abrogated the SAC in metaphase cells by adding an inhibitor of the SAC kinase Mps1, MPS1-IN (*Kwiatkowski et al., 2010*), 2:1 cells showed a bimodal distribution: 53% entered anaphase with symmetric spindles, with a median R centered around 1, however, the other 47% had asymmetric spindles with a distribution centered around R = 1.15 (*Figure 2H*, Mann–Whitney U test, p = 0.032 when compared to untreated 2:1 cells at anaphase onset). We conclude that the centering of the metaphase plate is not a mechanical consequence of cells preparing for anaphase, but that it depends on the SAC, which provides 2:1 cells with sufficient time to build a symmetric spindle. Mps1 inhibition in metaphase also led to a higher rate of segregation errors in 2:1 than in 2:2 or wild-type cells; this increase was reproducible, but not statistically significant (*Figure 2I,J*).

## An asymmetric distribution of centrioles leads to an imbalance of microtubule stability

To understand the mechanism behind metaphase plate centering, we investigated how the loss of a daughter centriole affects the spindle poles and why it may affect the position of the metaphase plate. We first tested whether loss of centrioles reduces the polar ejection force (*Rieder et al., 1986*), as a possible explanation for the shorter half-spindle attached to the pole with fewer centrioles. Wild-type eGFP-centrin1/mRFP–α-tubulin cells were compared with cells treated with RNAis against Sas-6 or the chromokinesin Kid, the main driving force of the polar ejection force (*Wandke et al., 2012*). Cells were subjected to a half-hour treatment with the Eg5 inhibitor monastrol to obtain monopolar spindles (*Mayer et al., 1999*), fixed and stained for HEC1 to label kinetochores. The average distance between kinetochores and centrosomes in monopolar spindles reflects the polar ejection force (*Wandke et al., 2012*), as seen by a reduced distance in Kid-depleted cells (*Figure 3A*). Loss of 2 daughter centrioles in Sas-6-depleted cells (one in each pole), however, did not reduce this distance, indicating that daughter centriole loss does not reduce the polar ejection force (*Figure 3A*). In *C. elegans*, half-spindle size has been linked to the abundance of TPX-2, an activator of Aurora-A, a critical regulator of centrosome maturation (*Nigg and Raff, 2009*; *Greenan et al., 2010*); however, by quantitative immunofluorescence we found no significant differences for TPX-2 or for phosphorylated Aurora-A (the active form of Aurora-A) between the two spindle poles in Sas-6-depleted 2:1 cells (*Figure 3B*). Next, we quantified the levels of centrobin, which stabilizes microtubule minus-ends, since it localizes only to daughter centrioles (*Zou et al., 2005*; *Jeffery et al., 2010*). Consistent with the daughter-specific localization,

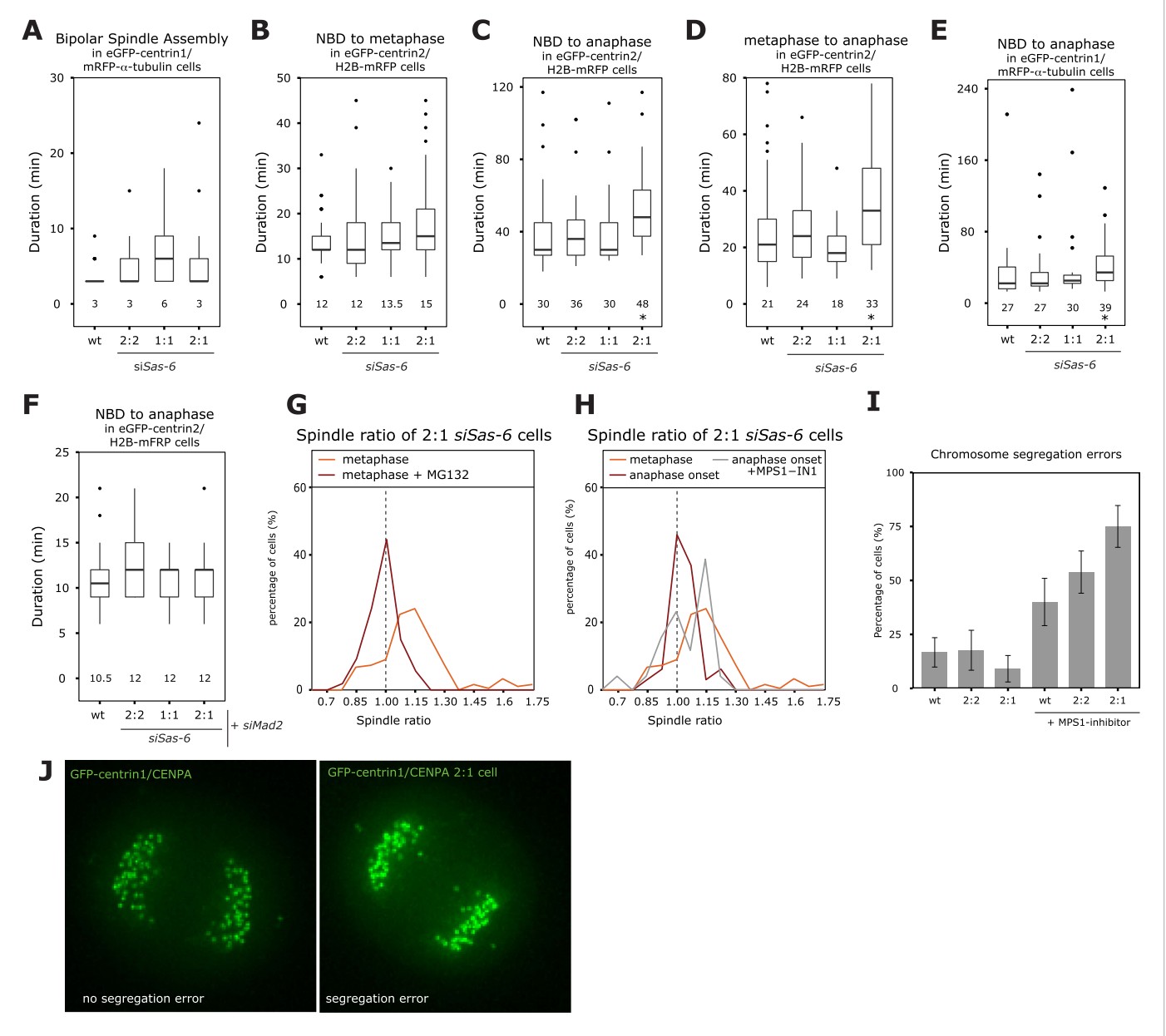

**Figure 2**. The SAC delays anaphase in cells with asymmetric spindles allowing the centering of the metaphase plate. (**A**) Boxplots of the spindle assembly time (NBD until bipolar spindle formation) in wild-type and Sas-6-depleted 2:2, 1:1, or 2:1 eGFP-centrin1/mRFP–α-tubulin cells. Numbers indicate the median value, n = 49–59 cells in 2–6 experiments. (**B**–**D**) Boxplots for the time between NBD and metaphase (**B**); the time between NBD and anaphase onset (**C**); and the time between metaphase and anaphase (**D**) in wild-type and Sas-6-depleted 2:2, 1:1, or 2:1 eGFP1-centrin2/H2B-mRFP cells. * indicates statistically significant difference in **C** (Mann–Whitney U test, p = 0.003), and **D** (Mann–Whitney U test, p = 0.015), n = 36–100 cells in 6–13 experiments. (**E**) Boxplot for the time between NBD and anaphase B in wild-type and Sas-6-depleted 2:2, 1:1, or 2:1 eGFP1-centrin1/ mRFP–α-tubulin cells. * indicates statistically significant difference (Mann–Whitney U test, p = 0.003). (**F**) Boxplots for the time between NBD and anaphase onset in Mad2-depleted or Mad-2/Sas-6-depleted 2:2, 1:1 or 2:1 eGFP1-centrin2/H2B-mRFP cells. 2:1 cells are not delayed (Mann–Whitney U test, p = 0.836). (**G**) Distribution of spindle ratios R in 2:1 eGFP-centrin1/CENPA cells treated with DMSO or MG132. For cell numbers see *Table 1*. (**H**) Distribution of spindle ratio R in Sas-6-depleted eGFP-centrin1/CENPA 2:1 cells in metaphase, at anaphase onset, or at anaphase onset when treated in metaphase with the Mps1 inhibitor MPS1-IN-1. Values for metaphase and anaphase without Mps1-IN treatment were taken from *Figure 1E* for comparison. MPS1-IN treated 2:1 anaphase cells are significantly more asymmetric (Mann–Whitney U test, p = 0.032). (**I**) Quantification of chromosome segregation errors in eGFP-centrin1/CENPA cells under the indicated conditions (n = 17–30 cells; N = 4–13 experiments). Error bars indicate s.e.m. (**J**) Illustrative live-cell imaging stills of eGFP-centrin1/CENPA cells in anaphase with (right panel) or without (left panel) chromosome segregation errors. SAC, spindle assembly checkpoint.

*Figure 2. continued on next page*

*Figure 2. Continued*

The following figure supplements are available for figure 2:

**Figure supplement 1**. Validation of Sas-6 and Mad2 co-depletion.

**Figure supplement 2**. Mps1 inhibition suppresses the anaphase delay in 2:1 cells.

centrobin was only present on the pole with two centrioles in 2:1 cells; in contrast, α-tubulin density in each half-spindle and the abundance of the centrosome proteins γ-tubulin, pericentrin, ninein, and p150$^{glued}$ (a marker for the dynein/dynactin motor complex) on the two spindle poles in 2:1 cells were as symmetric as in wild-type or 2:2 cells (*Figure 3C–F*). Since centrobin depletion leads to a SAC-dependent mitotic arrest due to unstable microtubule minus-ends and unstable kinetochore–microtubules (*Jeffery et al., 2010*), we hypothesized that the asymmetric localization of centrobin in 2:1 cells may lead to a difference in microtubule stability between the two poles. To test this, we subjected 2:2, 2:1, and 1:1 eGFP-centrin1/CENPA cells to a 7-min cold treatment on ice to depolymerize the astral microtubules and stained the remaining cold-resistant kinetochore–microtubules (*Salmon and Begg, 1980*). The stability of kinetochore–microtubules varied from cell to cell, but in 75% of the 2:2 or 1:1 cells the microtubule minus-ends showed the same stability at both spindle poles (*Figure 3G,H*). In contrast, in 2:1 cells, an unequal stability of the minus-ends was visible in over 50% of the cells, and in most cases it was the pole with 2 centrioles that had more stable minus-ends (*Figure 3G,H*). This difference in kinetochore–microtubule minus-end stability suggested that microtubule minus-ends might depolymerize faster in the absence of a daughter centriole, which would explain the shorter half-spindles associated with the 1-centriole pole. We also noted in our live-cell imaging experiments that >70% of the spindles in 2:1 eGFP-centrin1/mRFP-α-tubulin cells (but not 2:2 or 1:1 cells) were rotating, a phenomenon that could reflect a difference in minus-end stability of the two astral microtubule populations, resulting in an imbalance of cortical forces in the two half-spindles (*Figure 3I* and *Videos 1–3*). To confirm this hypothesis, we first tested whether a 1-hr MG132 treatment, which leads to symmetric spindles in 2:1 cells, is accompanied by an equalization in microtubule stability at the two spindle poles. While in control-treated cells, we saw an unequal stability of the minus-ends in $49 \pm 4\%$ of the 2:1 cells, this was only the case in $22 \pm 6\%$ of the MG132-treated 2:1 cells ($p = 0.0275$ in unpaired t-test; *Figure 3J*). In contrast, in 2:2 cells, MG132 treatment did not affect the percentage of cells with unequal minus-end stability (14% each; *Figure 3J*). Next, we tested whether the asymmetry of the spindle and the spindle rotation phenotype could be suppressed by increasing microtubule stability. We therefore co-depleted the microtubule-depolymerases KIF2a and MCAK, a condition known to increase microtubule stability (*Ganem and Compton, 2004*). KIF2a/MCAK depletion per se did not affect the position of the metaphase plate (median R = 1.01) or the ratio of cells with rotating spindles (*Figure 3K,L*). However, when KIF2a/MCAK was co-depleted in 2:1 cells, the spindle asymmetry was statistically less pronounced (median spindle ratio R = 1.08 and 19.2% R > 1.15 compared to median R = 1.12 and 42.9% R > 1.15 in Sas-6 deplete cells, $p = 0.039$ Mann–Whitney test), and spindles did not rotate (*Figure 3L,M*). To corroborate these results, we also stabilized microtubules by applying a brief, 30 min, 10 nM taxol treatment to Sas-6-depleted cells. While control (DMSO)-treated 2:1 cells were still asymmetric (median R = 1.23 and 56.3% R > 1.15), taxol-treated cells were much closer to symmetry (median R = 1.02 and 13.3% R > 1.15, $p = 0.0062$; *Figure 3N*). We conclude that loss of a single centriole leads to a difference in minus-end microtubule stability between the two spindle poles, that this difference plays a large role in the asymmetric positioning of the metaphase plate, and that cells reduce this difference as they center the metaphase plate before anaphase onset.

## 2:1 cells form amphitelic kinetochore–microtubule attachments that take longer to stabilize

In the next step, we investigated whether the difference in minus-end stability in 2:1 cells translated into kinetochore–microtubule attachment defects sensed by the SAC. The SAC responds to unattached kinetochores or to sister-kinetochores with insufficient tension (measured as the distance

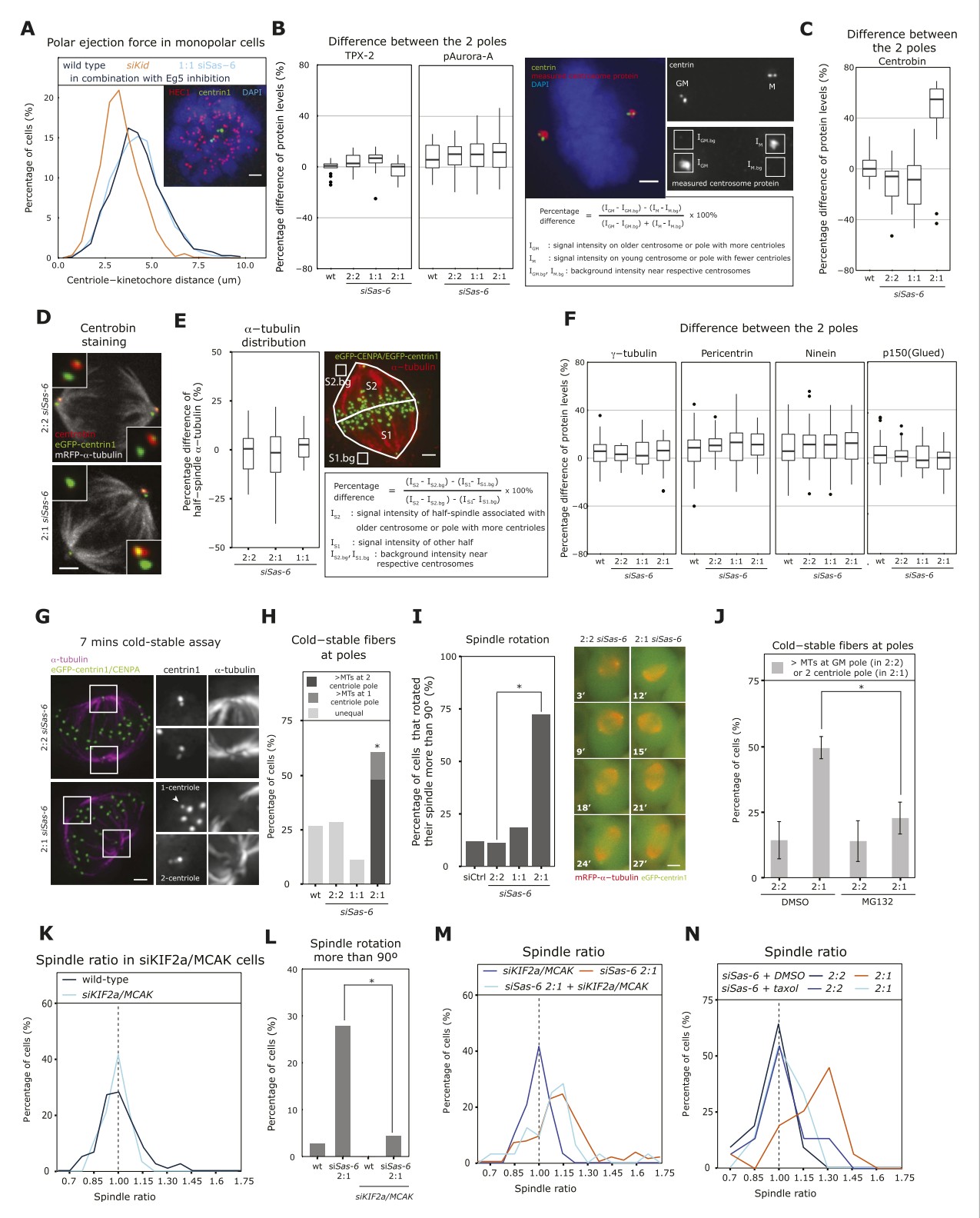

**Figure 3**. 2:1 cells have half-spindles with different microtubule stability and fail to mature kinetochore–microtubule attachments. (**A**) eGFP-centrin1 cells (green) were treated with monastrol and stained for HEC1 (red; right panel) to calculate the distance between kinetochores and the closest centriole. The left panel shows the distribution of centriole-HEC1 distances from n =16–28 cells, >1000 kinetochores. Scale bar indicates 2 μm. (**B**, **C**) The difference in centrosomal levels of TPX-2 and phospho-Aurora-A (**B**), and centrobin between each spindle pole was quantified in HeLa eGFP-centrin1 by

*Figure 3. continued on next page*

*Figure 3. Continued*

immunofluorescence using the indicated formula, and plotted as boxplots for each centriole configuration; n = 12–63 cells. Scale bar indicates 2 µm. (**D**) Representative image of wild-type and *siSas-6*-treated 2:1 eGFP-centrin1 (green)/mRFP–α-tubulin (white) cells stained with anti-centrobin sera (red). Insets show magnified centrioles. Scale bar indicates 2 µm. (**E**) Quantification of the difference in α-tubulin (red signal) levels between the two half-spindles, according to the formula shown in the box. Results were plotted in the left panel using a boxplot. n = 19–25 cells, N = 2 experiments. Scale bar indicates 2 µm. (**F**) The difference in centrosomal levels of γ-tubulin, pericentrin, ninein and p150$^{glued}$ between each spindle pole was quantified as in (**B**) and plotted as boxplots for each centriole configuration; n = 18–68 cells. (**G**) Immunofluorescence images of 2:2 and 2:1 *siSas-6* eGFP-centrin1/CENPA (green) cells treated for 7 min with ice-cold medium and stained with anti-α-tubulin sera (magenta). Subsetted images are maximum intensity projections of 10 stacks (z = 0.2 µm) around centrioles. Scale bar indicates 2 µm. (**H**) Quantification of kinetochore–microtubule minus-end stability at poles. Bar graph indicates percentage of cells that have asymmetric levels of kinetochore–microtubule minus-ends at the poles after cold-treatment; n = 26–49 cells; * indicates that within the 2:1 cell population the minus-end stability was significantly higher at the pole with 2 centriole (p = 0.00082 exact binomial test). (**I**) Quantification of spindle rotation in control- and Sas-6-depleted 2:2, 2:1, or 1:1 eGFP-centrin1 (green)/mRFP–α-tubulin (red) cells based on time-lapse images as shown in the right panels. Times indicate minutes after NBD. Scale bar indicates 5 µm. A spindle was counted as rotating if it had turned by more than 90° in X/Y. n = 32–122 cells in 2–6 experiments. * indicates significant difference; Fisher's exact test p = 8.39e-09. (**J**) Quantification of kinetochore–microtubule stability at poles. Bar graph indicates percentage of cells that have more stable kinetochore–microtubule minus-ends either at the pole with the grandmother centriole (2:2) cells or at the 2-centriole pole (2:1 cells); n = 20–40 cells in N = 3 independent experiments; * indicates that the MG132 treatment significantly reduced the percentage of cells with more stable minus-ends at the 2-centriole pole (p = 0.0275 in unpaired t-test). (**K**) Distribution of spindle ratios R in wild-type and siKIF2a/MCAK-treated HeLa eGFP-centrin1/CENPA cells in metaphase. For cell number see *Table 1*. (**L**) Quantification of spindle rotation in wild type, Sas-6-depleted 2:1, KIF2a/MCAK-depleted, or KIF2a/MCAK/Sas-6-depleted 2:1 eGFP-centrin2/H2B-mRFP cells, n = 18–37 cells in 1–3 experiments. * indicates significant difference; p = 0.024 in Fisher's exact test. See also *Videos 1–3*. (**M**) Distribution of spindle ratios R in Sas-6-depleted 2:1 and KIF2a/MCAK/Sas-6-depleted 2:1 eGFP-centrin1/CENPA cells in metaphase. For cell numbers see *Table 1*. (**N**) Distribution of spindle ratios R in metaphase in Sas-6-depleted 2:2 and 2:1 cells treated either with DMSO or 10 nM taxol. For cell numbers see *Table 1*.

between the two sister-kinetochores) that become transiently detached due to the kinase activity of Aurora-B (*Foley and Kapoor, 2013*). Depletion of centrobin on both spindle poles leads to unstable kinetochore–microtubules and reduced inter-kinetochore distances that result in a permanent mitotic arrest (*Jeffery et al., 2010*). When we tracked kinetochores in 2:1 eGFP-centrin1/CENPA cells, however, we found no difference in inter-kinetochore distances or the oscillatory sister-kinetochore movements along the spindle axis (*Figure 4A,B*; *Jaqaman et al., 2010*). Since 42.9% of the 2:1 cells have asymmetric spindles, we restricted our analysis to only include cells that have a spindle ratio R > 1.15, but again found no change in inter-kinetochore distances (*Figure 4A*). As SAC satisfaction is also linked to intra-kinetochore stretching (*Maresca and Salmon, 2009*; *Uchida et al., 2009*), we further measured the intra-kinetochore distance between CENPA and the outer kinetochore protein HEC1 and found no change in 2:1 cells compared to 2:2 or 1:1 cells (*Figure 4C*); the observed distance of 110 nm is consistent with previous studies (*Wan et al., 2009*). Overall, this indicated that sister-kinetochores in 2:1 cells formed bipolar attachments with mechanical behaviors that were indistinguishable from normal cells. Since a single unattached kinetochore is sufficient to elicit a SAC response (*Rieder et al., 1995*; *Collin et al., 2013*), we next investigated whether 2:1 cells have rare unattached kinetochores, by measuring the proportion of metaphase cells with Mad2-positive kinetochores (marker for unattached kinetochores). When compared to 2:2, 1:1, or wild-type cells, we found a small but non-significant increase in the proportion of Mad2-positive 2:1 cells (*Figure 4D*), which could point to a minor or transient attachment defect caused by the imbalance of microtubule stability within the spindles of 2:1 cells. Such imbalance could lead to kinetochores that are not fully attached, that is, not bound by the full complement of microtubules (up to 25 microtubules per kinetochore in vertebrate cells; *Rieder, 1982*). To detect immature attachments, metaphase eGFPcentrin1/CENPA cells were stained with antibodies against SKAP, a marker for kinetochores fully bound by stable microtubules (*Schmidt et al., 2010*), and analyzed for the presence of SKAP on kinetochores: cells in which more than 30% of the kinetochore-pairs were SKAP-negative were considered to have partially unstable kinetochore–microtubules (*Figure 4E,F*). Such cells were rare in wild-type, 2:2, or 1:1 cells (<10%), but formed a substantial proportion of 2:1 cells (22%; *Figure 4F*). These SKAP-negative kinetochores were distributed in a symmetric manner (were present both on the 1- or 2-centriole side; data not shown), and they all disappeared if 2:1 cells were briefly treated with 10 nmol taxol (*Figure 4F*). Moreover a 1-hr treatment with the proteasome inhibitor MG132 also abolished the subpopulation of 2:1 cells with SKAP-negative kinetochores (*Figure 4F*). Overall this suggested that in 2:1 cells, sister-kinetochores form amphitelic microtubule attachments, but that

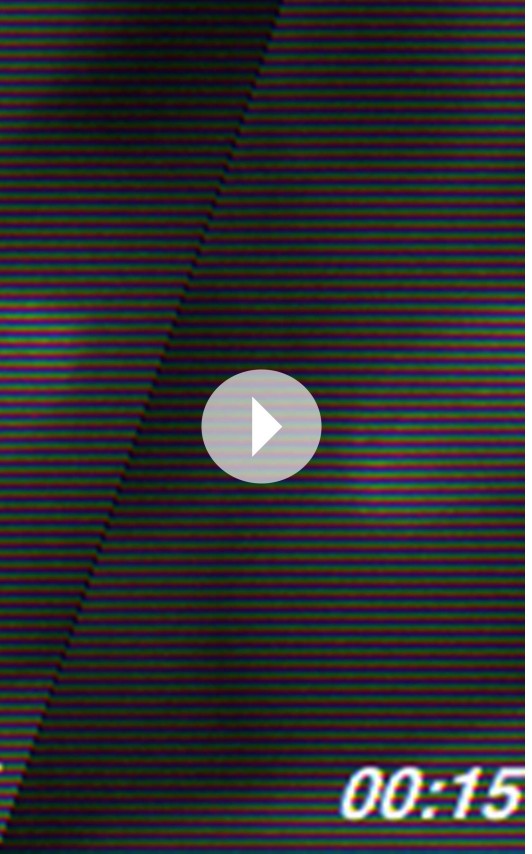

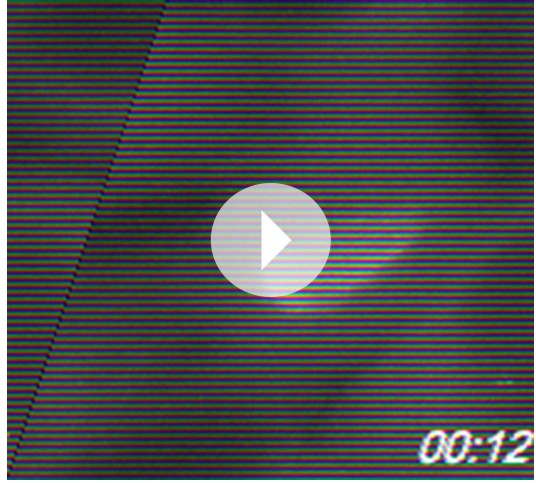

**Video 2.** Sas-6-depleted 2:1 HeLa cell expressing eGFP-centrin1 (centriole marker; green) and mRFP-α-tubulin (microtubules; red) in mitosis. Time is indicated in minutes. Note the spindle rotation movements.

**Video 1.** Sas-6-depleted 2:2 HeLa cell expressing eGFP-centrin1 (centriole marker; green) and mRFP-α-tubulin (microtubules; red) in mitosis. Time is indicated in minutes.

differences in minus-end stability delay stabilization of kinetochore–microtubule attachments in both half-spindles, as reflected by higher levels of SKAP-negative kinetochores.

Since our results contrasted with recent hypotheses that suggest that correct force generation at kinetochores is locally regulated and does not require a direct connection between kinetochores and centrosomes (*Sikirzhytski et al., 2014*), we aimed to exclude a possible off-target effect of Sas-6 depletion on kinetochore–microtubule attachments. For this purpose, we used laser-microsurgery to ablate one daughter centriole in cells that had already reached metaphase. As in Sas-6-depleted cells, 2:1 cells generated by microsurgery resulted in an asymmetric plate location, as, on average, R increased from 1.06 to 1.16 after laser-ablation (*Figure 4G,H* and *Video 4*). These cells displayed a reduced inter-kinetochore distance (mean of 1.03 μm before ablation to 0.95 μm after the ablation, p = 0.001 in paired two-tailed t-test), which led to a prolonged metaphase arrest (*Figure 4I*); in contrast, a control laser pulse in the vicinity of the spindle pole did not affect half-spindle lengths or inter-kinetochore distances (data not shown). This confirmed that loss of a single daughter centriole can directly affect force generation at sister-kinetochores, consistent with our findings that Sas-6-depleted 2:1 cells show defects in the quality of kinetochore–microtubule attachments. We also conclude that an acute laser-ablation of a daughter centriole in metaphase has a more severe effect on the forces acting on kinetochore than Sas-6 depletion. Possible explanations for this difference could be that Sas-6-depleted 2:1 cells might have more time to adapt to the lack of a missing centriole as they progressively build up the spindle, or that the laser-ablation destroys not just a centriole but also part of enzymatic activities in the vicinity of the centriole, such as minus-end depolymerases, which in normal cells are known to exert a pulling force on kinetochore-fibers (*Meunier and Vernos, 2011*).

## Stabilization of kinetochore–microtubules suppresses the SAC-dependent delay in 2:1 cells

To confirm the presence of partially destabilized kinetochore–microtubule attachments in *siSas-6*-treated 2:1 cells and test whether they are the cause of the SAC response, we used the Aurora-B

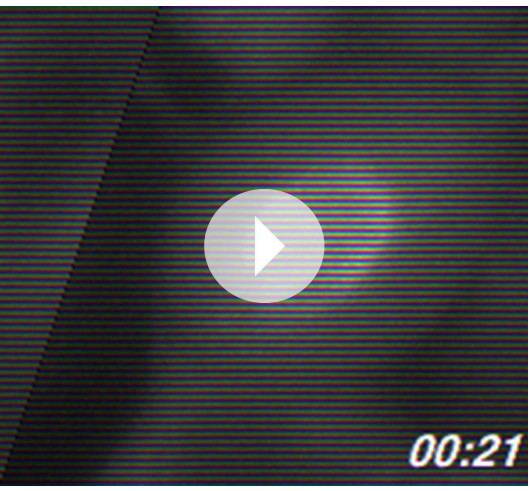

**Video 3.** Sas-6-depleted 1:1 HeLa cell expressing eGFP-centrin1 (centriole marker; green) and mRFP-α-tubulin (microtubules; red) in mitosis. Time is indicated in minutes.

inhibitor ZM1. Aurora-B inhibition does not overcome a SAC-dependent mitotic arrest caused by unattached kinetochores (nocodazole treatment), but overcomes a mitotic arrest caused by insufficient tension at sister-kinetochores in monopolar spindles (monastrol treatment), as Aurora-B inhibition stabilizes kinetochore–microtubules and prevents loss of kinetochore–microtubule attachment (*Figure 5A*; *Ditchfield et al., 2003*; *Lampson et al., 2004*). When we acutely inhibited Aurora-B in metaphase 2:1 cells, cells entered anaphase with asymmetric spindles (median R = 1.18, Mann–Whitney U test, p = 7.57 × 10$^{-6}$ when compared to anaphase onset without Aurora-B inhibitor), indicating that a stabilization of kinetochore–microtubules satisfies the SAC in 2:1 cells (*Figure 5B*). To confirm these findings in an independent manner, we also tested whether depletion of the microtubule depolymerases KIF2a/MCAK would allow 2:1 cells to enter anaphase with asymmetric spindles. Depletion of KIF2a/MCAK per se did not affect plate position at anaphase onset (median R = 1.03); however, it allowed 2:1 cells to enter anaphase with asymmetric spindles (median R 1.08, p < 0.0001 compared to symmetric distribution in Mann–Whitney U test; *Figure 5C*). KIF2a/MCAK depletion also rescued the anaphase timing difference in 2:1 cells compared to 2:2 cells and led to the correct loading of SKAP on kinetochores (*Figure 5D,E*; note that KIF2a/MCAK depletion alone led to a small anaphase delay when compared to wild-type cells, as previously reported [*Ganem et al., 2005*]). This showed that co-depletion of KIF2a/MCAK overrides the SAC response in 2:1 cells, leaving them no time to center the metaphase plate, indicating that unstable kinetochore–microtubules cause the SAC-dependent anaphase delay. KIF2a/MCAK co-depletion, however, did not suppress a mitotic arrest caused by the presence of insufficient tension (monastrol treatment) or unattached kinetochores (nocodazole treatment; *Figure 5A*). Since the spindle checkpoint response has been shown to be subject to off-target effects (*Hübner et al., 2009*; *Westhorpe et al., 2010*), we repeated these experiments with an alternative set of KIF2a and MCAK siRNAs. These experiments validated our initial findings, confirming that KIF2a/MCAK suppress the anaphase delay seen in Sas-6-depleted 2:1 cells (*Figure 5—figure supplement 1A*). Validation of Sas-6, KIF2a, and MCAK depletion siRNA treatments by immunoblotting also showed that in the triple MCAK/KIF2a/Sas-6 depletion, KIF2a was only partially depleted (*Figure 5—figure supplement 1B*). We therefore tested whether the depletion of MCAK alone in 2:1 cells is sufficient to suppress the mitotic delay. As this was not the case, we conclude that depletion of both MT-depolymerases is necessary to overcome the instability of kinetochore–microtubules in 2:1 cells (*Figure 5—figure supplement 1C*).

## Anaphase entry with asymmetric spindles leads to asymmetric cell division

Since the mere depletion of KIF2a/MCAK allowed 2:1 cells to enter anaphase with asymmetric spindles, we could investigate the functional importance of a centered metaphase plate position beyond anaphase onset (Aurora-B inhibition could not be used, since it blocks anaphase B and cytokinesis [*Ditchfield et al., 2003*]). We found that KIF2a/MCAK/Sas-6-depleted 2:1 cells had slightly higher rates of chromosome segregation errors (statistically insignificant difference, p = 0.4713, Fischer's exact test) when compared to KIF2a/MCAK-depleted cells (*Figure 6A*). Strikingly KIF2a/MCAK-depleted 2:1 cells also underwent asymmetric cell divisions, yielding two daughter cells of different volumes (*Figure 6B* and *Videos 5, 6*): while the two daughter cell volumes never differed by more than 20% in wild-type or KIF2a/MCAK depleted cells, 50% of KIF2a/MCAK/Sas-6-depleted 2:1

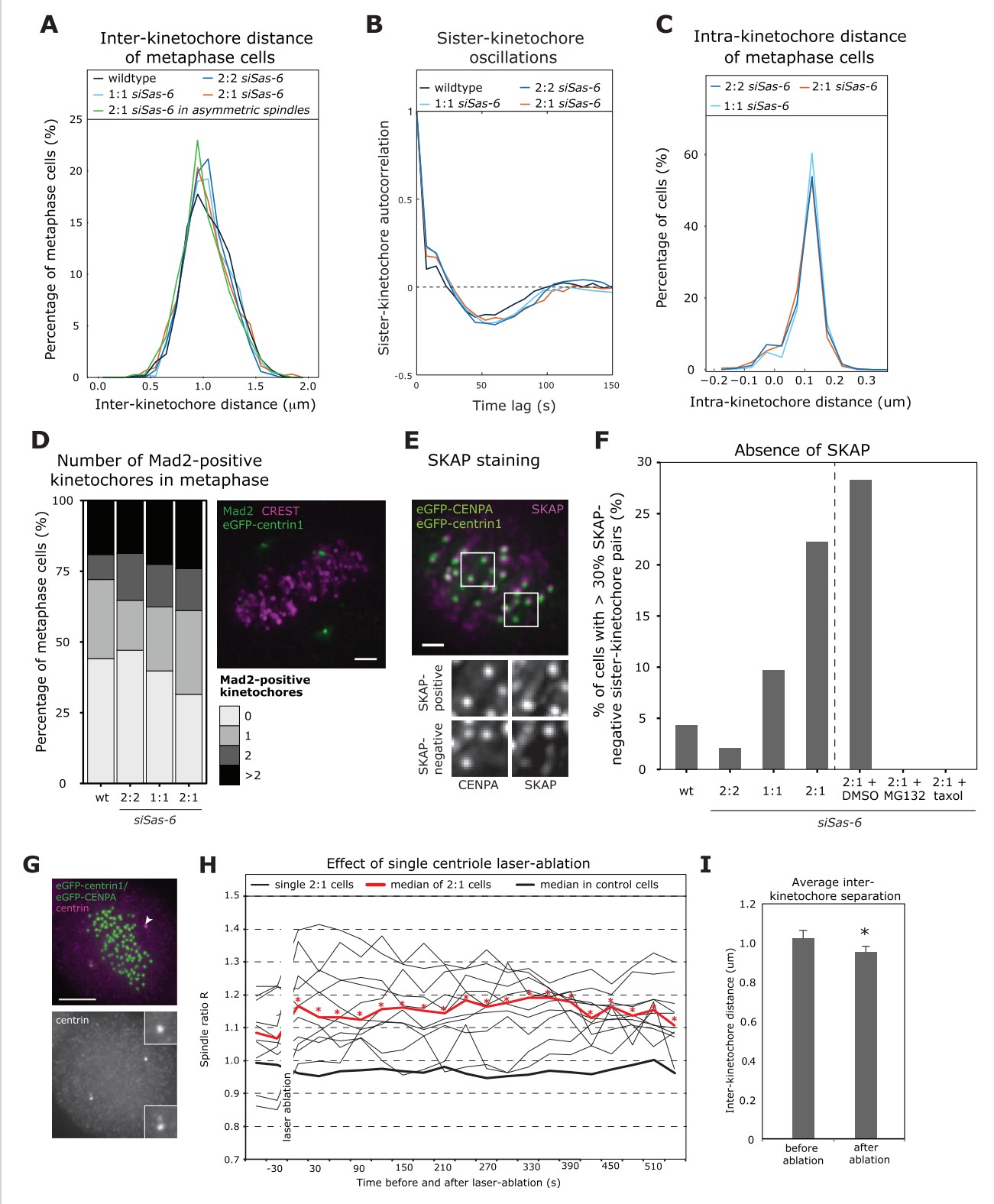

**Figure 4**. 2:1 cells have immature kinetochore–microtubule attachments. (**A**, **B**) Analysis of inter-kinetochore distances and sister-kinetochore oscillations in wild-type, Sas-6-depleted 2:2, 1:1, 2:1, or the subset of 2:1 eGFP1-centrin1/CENPA metaphase cells with an asymmetric plate position based on our in-house kinetochore tracking assay (*Jaqaman et al., 2010*), n = 620–889 kinetochores in 36–48 cells. The distribution of inter-kinetochore distances (CENPA to CENPA distance) is shown in **A** (no significant difference, t-test, p = 0.99), and the autocorrelation of the sister-kinetochore movements in **B**. The first

*Figure 4. continued on next page*

*Figure 4. Continued*

minima of the autocorrelation curve indicate the half-period of the chromosome oscillations, and their depth the regularity of the oscillations. (**C**) Distribution of intra-kinetochore distances in wild-type and Sas-6-depleted 2:2, 1:1, or 2:1 eGFP1-centrin1/CENPA metaphase cells. Cells were stained with antibodies against the N-terminus of HEC1. Using the tracking assay, we determined for each sister-kinetochore pair the CENPA–CENPA and the HEC1-HEC1 distances, and calculated the CENPA-HEC1 distances by halving the difference, n = 701–790 kinetochores in 26–30 cells in 3 experiments (no significant difference, Mann–Whitney test, 2:1 vs 2:2, p = 0.203). (**D**) Quantification of Mad2-positive kinetochores in wild-type or Sas-6-depleted 2:2, 1:1, or 2:1 cells. eGFP-centrin1 (green) metaphase cells were stained with anti-Mad2 (green), and CREST sera (magenta; left panel) and the number of Mad2-positive kinetochores quantified in the right panel (n = 50–102 cells in 2 (wt) or 8 (siSas-6) experiments; no significant difference was found; Fisher's exact test, p = 0.17). Scale bar indicates 2 μm. (**E**, **F**) Quantification of SKAP-negative kinetochores in wild-type, siSas-6-depleted 2:2, 2:1, 2:1 + DMSO, 2:1, 2:1 + MG132, and 2:1 + taxol-treated eGFP-centrin1/CENPA cells. Cells were stained with antibodies against the kinetochore protein SKAP (magenta), as shown in **E** (maximum-intensity projection of 8 stacks [z = 0.3 μm]). Using the eGFP-CENPA (green) signal, we quantified the number of sister-kinetochore pairs with at least one SKAP-negative kinetochore (as shown in inlets). Quantification in **F** shows the percentage of cells where more than 30% of the sister-pairs were SKAP-negative. Fisher's exact test for 2:1 > 2:2, p = 0.0013. n = 46–72 cells in N = 4–7 experiments. Scale bar indicates 2 μm. (**G**) Example of an eGFP-centrin1/CENPA (green) cell in which a single daughter centriole was ablated (white arrow indicates the location of the laser pulse). Cells were fixed and stained with anti-centrin sera (magenta) to confirm the loss of a centriole, as opposed to the mere bleaching of eGFP-centrin1. Scale bar indicates 5 μm. (**H**) Plot of half-spindle ratio R over time in 11 single eGFP-centrin1/CENPA cells in which a single centriole was ablated. The time point of laser ablation is t = 0. The thick red curve indicates the median of R of laser ablated 2:1 cells (* denotes when median R is asymmetric [p < 0.01]), the thick black curve indicates the median R distribution of 8 control-ablated cells. (**I**) Average inter-kinetochore distances in eGFP-centrin1/CENPA cell before or after a single daughter centriole was ablated as determined by the kinetochore tracking assay. Error bars indicate s.e.m. n = 11 cells, 2 time points before and after ablation and on average 20 kinetochores per cell. * denotes a statistically significant difference (p = 0.001 in two-tailed paired t-test).

cells yielded one daughter cell that was at least 20% larger than its sister progeny (*Figure 6C*). This suggested that an asymmetric metaphase plate position leads to asymmetric cell division. A recent study, however, demonstrated that asymmetric position of the entire spindle can also lead to asymmetric cell divisions in HeLa cells (*Kiyomitsu and Cheeseman, 2013*). To discriminate between the two possibilities, we quantified the position of the spindle center in relationship to the cell center at anaphase onset, which was often not centered in the middle of the cell at anaphase onset, as has been previously reported (*Figure 6D*; *Collins et al., 2012*; *Kiyomitsu and Cheeseman, 2013*). The extent to which the spindle centers were offset was the same in KIF2a/MCAK-depleted and KIF2a/MCAK-depleted 2:1 cells, indicating that changes in spindle position were not at the origin of the asymmetric cell divisions seen in KIF2a/MCAK-depleted 2:1 cells. To independently confirm our hypothesis that an asymmetric metaphase plate position in anaphase leads to an asymmetric cell division, we acutely abrogated the spindle checkpoint with an Mps1 inhibitor in control-depleted cells, Sas-6-depleted 2:2 cells, Sas-6-depleted 2:1 cells in late prometaphase/early metaphase (as visualized by live-cell imaging, where 43% of the cells have a metaphase plate not in the middle of the spindle, see *Figure 1F*) or Sas-6-depleted 2:1 cells that were in late metaphase, which have symmetric spindles. Mps1 inhibition did not give rise to asymmetric cell division in control-depleted cells, 2:2 cells, or 2:1 cells treated in late metaphase; in contrast, in late prometaphase/early metaphase 2:1 cells with an asymmetric plate position, Mps1 inhibition resulted in an asymmetric cell division in 45% of the cases (*Figure 6E*). The same result was also seen after acute Mps1 inhibition in laser-ablated cells: cells that were treated with a laser

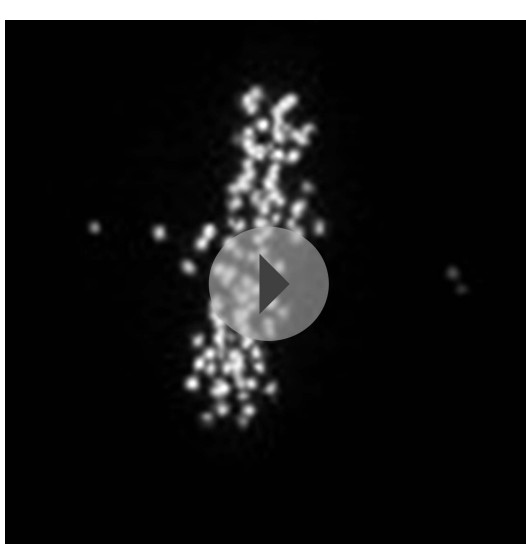

**Video 4.** Laser-ablated 2:1 HeLa cell expressing eGFP-centrin1 (centriole marker) and eGFP-CENPA (kinetochore marker) in metaphase. Note the asymmetric metaphase plate position after the ablation of a single centriole.

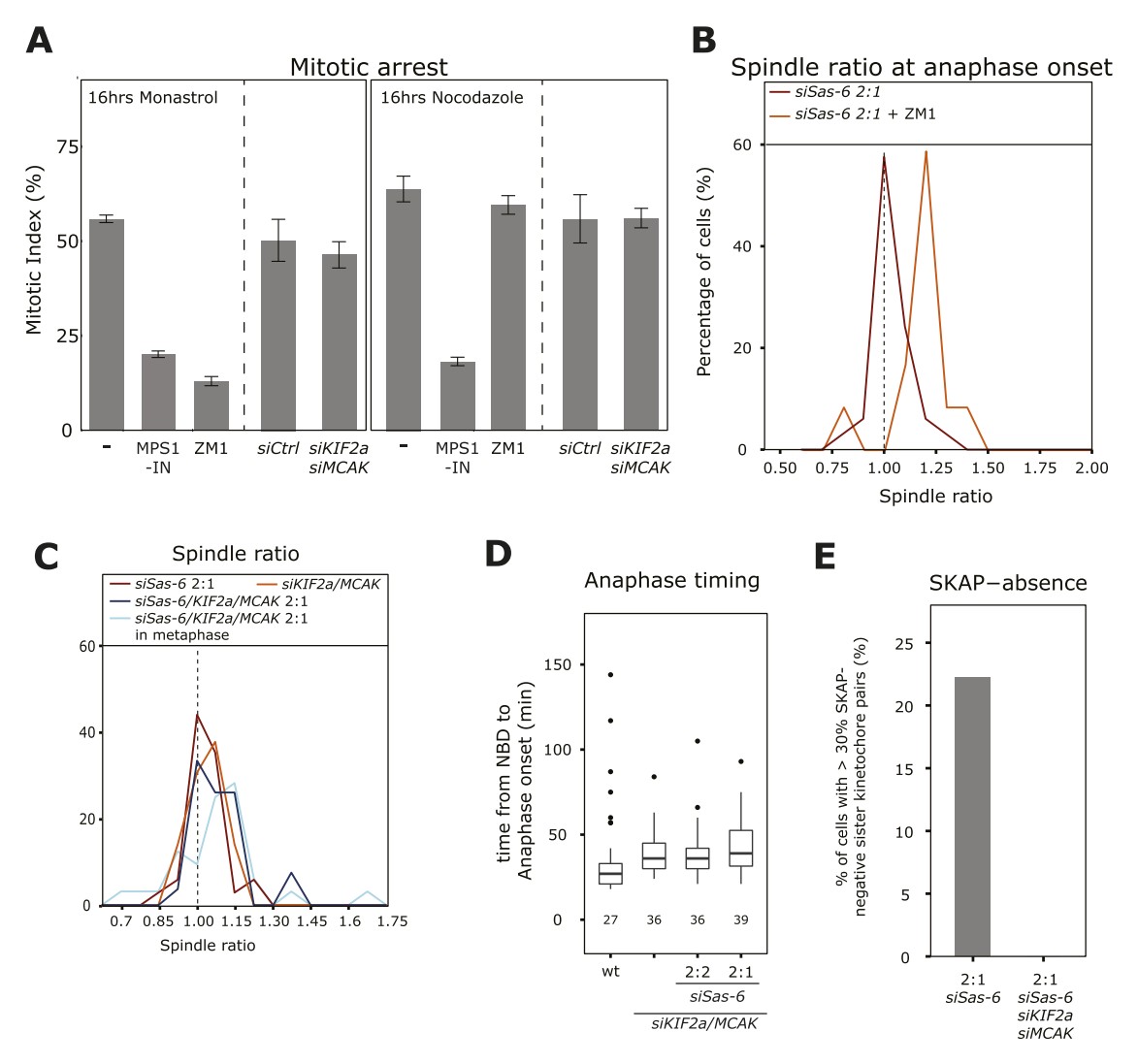

**Figure 5**. Depleting KIF2a and MCAK overcomes the SAC response in 2:1 cells. (**A**) Mitotic index of untreated, ZM-1-treated, MPS1-IN-treated, control-depleted, or KIF2a/MCAK-depleted cells treated for 16 hr with nocodazole (unattached kinetochores) or monastrol (lack of tension), n ≥ 400 cells in 3–4 experiments, error bars indicate s.e.m. * ZM1 and MPS1-IN overcome a monastrol arrest (t-test p < 0.0001), and MPS1-IN overcomes a nocodazole arrest (t-test p = 0.0044). (**B**) Distribution of spindle ratio R in Sas-6-depleted eGFP-centrin1/CENPA 2:1 cells at anaphase onset treated with or without the Aurora-B inhibitor ZM1. Data from *Figure 1E* without Aurora-B inhibition are shown for comparison. Aurora-B inhibition allows cells to enter anaphase with asymmetric spindles (n = 12 cells; Mann–Whitney U test, p = $7.57 \times 10^{-6}$). (**C**) Distribution of spindle ratios R in Sas-6-depleted 2:1, KIF2a/MCAK-depleted, and KIF2a/MCAK/Sas-6-depleted 2:1 eGFP-centrin1/CENPA cells in metaphase or at anaphase onset. (**D**) Boxplots of anaphase timing of wild-type, KIF2a/MCAK-depleted, or KIF2a/MCAK/Sas-6 depleted 2:2 and 2:1 eGFP1-centrin2/H2B-mRFP cells. n = 23–61 cells in 1–3 experiments.
(**E**) Quantification of SKAP-negative kinetochores as in *Figure 4F* in Sas-6-depleted 2:1 and KIF2a/MCAK/Sas-6-depleted 2:1 eGFP-centrin1/CENPA cells. SAC, spindle assembly checkpoint.
The following figure supplement is available for figure 5:

**Figure supplement 1**. Validation of Sas-6, KIF2a, and MCAK co-depletion.

pulse in the cytoplasm divided in a symmetric manner; in contrast, 67% of the cells in which a single centriole was laser-ablated divided in an asymmetric manner when treated with an Mps1 inhibitor (*Figure 6F* and supplementary *Videos 7, 8*). We conclude that entry into anaphase with an asymmetric position of the metaphase plate results in an asymmetric cell division.

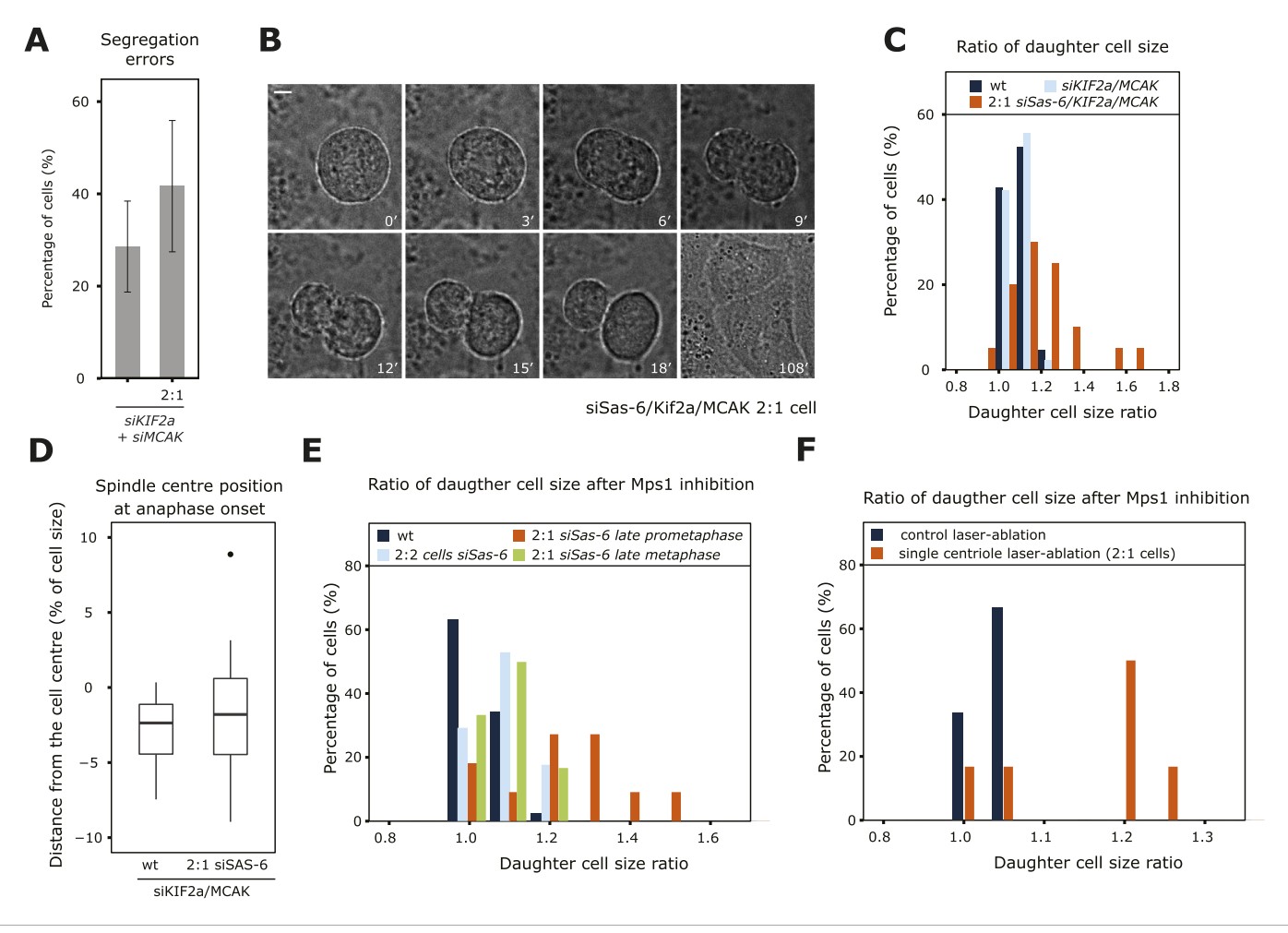

**Figure 6**. An asymmetric plate position at anaphase onset leads to segregation errors and asymmetric cell division. (**A**) Quantification of chromosome segregation errors in KIF2a/MCAK-depleted and KIF2a/MCAK/Sas-6-depleted 2:1 eGFP-centrin1/eGFP-CENPA cells based on time-lapse images. n = 12–21 cells in 2–4 experiments. Error bars indicate s.e.m. (**B**, **C**) Wild-type, KIF2a/MCAK-depleted or KIF2a/MCAK/Sas-6-depleted 2:1 eGFP-centrin1/ CENPA cells were recorded by time-lapse imaging using the eGFP-centrin1 signal to count centrioles and phase contrast to detect the cell membrane as shown in **B** for a siKIF2a/MCAK/Sas-6 2:1 cell (scale bar = 5 μm). Phase contrast images were used to quantify the ratio of the two daughter cell sizes, which was plotted as a histogram in **C**. Half the Kif2a/MCAK/Sas-6-depleted 2:1 cells had a ratio of over 1.2, a ratio never observed in other conditions (t-test with Welch's correction between KIF2a/MCAK and KIF2a/MCAK/Sas-6 2:1, p = 3.1e-05; n = 20–45 cells in 2–4 experiments). (**D**) Quantification of spindle center position in relation to cell center in KIF2a/MCAK-depleted or KIF2a/MCAK/Sas-6 2:1 eGFP-centrin1/CENPA cells at anaphase onset. n = 20–45 cells in 2–4 experiments. (**E**) Wild-type HeLa eGFP-centrin1/eGFP-CENPA cells, 2:2 cells, 2:1 cells in late prometaphase (still 1–2 chromosomes not perfectly aligned on the plate), or 2:1 cells in late metaphase (plate perfectly in the middle) were treated with an Mps1 inhibitor and recorded by time-lapse imaging using phase contrast to detect the cell membrane as shown in **B**. Shown is the ratio of the two daughter cell sizes; 45% of the 2:1 cells treated in late prometaphase/early metaphase had a ratio of over 1.2, a ratio never observed in other conditions (t-test with Welch's correction between 2:1 cells in late prometaphase and 2:1 cells in late metaphase, p = 0.0141; n = 11–17 cells in 3 experiments). See also *Videos 5, 6*. (**F**) HeLa eGFP-centrin1/CENPA cells were treated with a laser pulse in the cytoplasm (control) or ablation of a centriole (2:1 cells) and acutely treated with an Mps1 inhibitor to force cells into anaphase. Shown is the ratio of the two daughter cell sizes. Note that 67% of the 2:1 cells treated in late prometaphase/early metaphase had a ratio of over 1.2, a ratio never observed in other conditions (t-test with Welch's correction between control and 2:1 cells p = 0.0451; n = 6). See also *Videos 7, 8*.

## Discussion

Here, we show that an equatorial position of the metaphase plate in the middle of the spindle is necessary for symmetric cell divisions and demonstrate that cells actively center the metaphase plate before anaphase onset. Metaphase plate centering requires the SAC, which provides cells with enough time to correct metaphase plate position. The SAC responds to subtle defects in

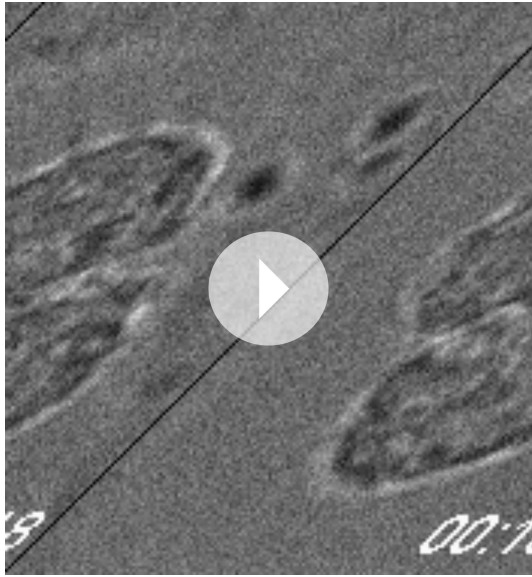

**Video 5.** KIF2a/MCAK-depleted HeLa cell expressing eGFP-centrin1 (centriole marker; green) and eGFP-CENPA (kinetochore marker; green) entering anaphase and recorded with phase contrast microscopy.

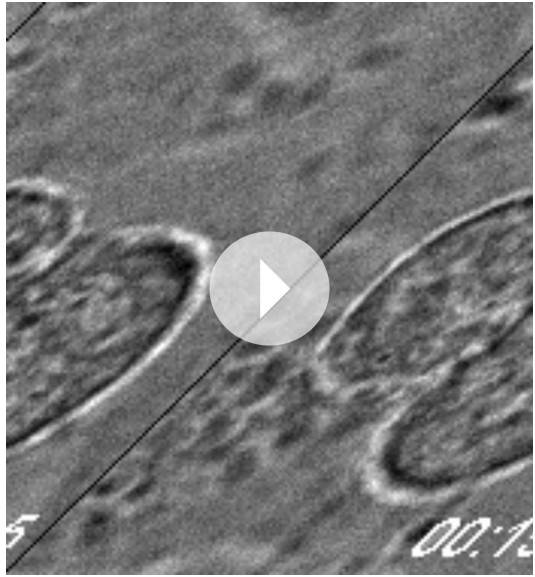

**Video 6.** KIF2a/MCAK/Sas-6-depleted 2:1 HeLa cell expressing eGFP-centrin1 (centriole marker; green) and eGFP-CENPA (kinetochore marker; green) entering anaphase and recorded with phase contrast microscopy. Note the asymmetric cell division.

kinetochore–microtubule stability that arise in cells with an asymmetric plate position and an imbalance of centrioles, implying that the SAC is more sensitive than previously assumed.

Recent studies have shown that proper positioning of the spindle ensures symmetric cell divisions, and that, deviations from a symmetric position are corrected by dynein-dependent cortical forces and membrane elongation during anaphase (*Kiyomitsu and Cheeseman, 2013*). Here, we find that this external cortical correction mechanism in anaphase is complemented in metaphase by an internal centering mechanism that ensures a symmetric position of the metaphase plate within the spindle with the help of the SAC. This centering mechanism is particularly visible in cells with an asymmetric distribution of centrioles (2:1 cells), but it also acts in wild-type cells, indicating that it is active in every cell division. The centering of the metaphase plate is not just a consequence of establishing stable bipolar attachments at kinetochores, since KIF2a/MCAK/Sas-6-depleted 2:1 cells fail to center the plate, despite having reached stable bipolar attachments that satisfy the SAC; it is an active correction process, which in part depends on the regulation of microtubule dynamics, but whose precise molecular mechanisms will need to be uncovered.

KIF2a/MCAK-depleted 2:1 cells fail to center the metaphase plate before anaphase and divide asymmetrically. These cells have no defect in spindle positioning when compared to the symmetrically-dividing KIF2a/MCAK-depleted cells, implying that the asymmetric position of the metaphase plate is the source of asymmetric cell division. This hypothesis is confirmed by our analysis of 2:1 cells treated with an Mps1 inhibitor: only 2:1 cells treated in late prometaphase/early metaphase, which still have asymmetrically positioned metaphase plates, give rise to asymmetric cell division, whereas the same 2:1 cell population treated in late metaphase, which have a centered metaphase plate position, carry out symmetric cell divisions. We thus postulate that a symmetric metaphase plate position is essential for symmetric cell divisions, explaining why it is conserved in all metazoans, plants, and many fungi. Control of this parameter is essential, since differences in cell size have been linked to cell fate (*Kiyomitsu and Cheeseman, 2013*). Metaphase plate position may also play a crucial role in asymmetric cell divisions that depend on asymmetric spindles in anaphase, such as in embryonic *D. melanogaster* neuroblasts. To form asymmetric spindles in a controlled and stereotypical manner, cells need an internal reference in space: breaking an existing symmetry, that is, a symmetric metaphase plate position, provides such

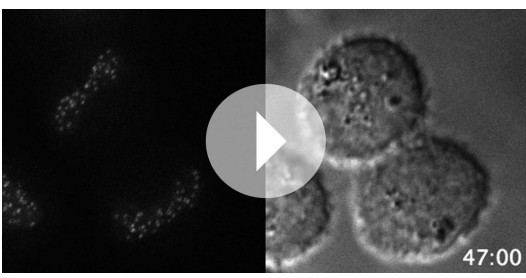

**Video 7.** Laser-ablated control (ablation in the cytoplasm) HeLa cell expressing eGFP-centrin1 (centriole marker) and eGFP-CENPA (kinetochore marker) treated in metaphase with an Mps1 inhibitor. Shown is the GFP-fluorescence channel (left) and the DIC channel (right). Note how the cell divides in a symmetric manner.

a reference point. This is consistent with the progression of embryonic fly neuroblasts, which first align the metaphase plate in the middle of the spindle, before undergoing an asymmetric elongation of the spindle in anaphase. Our results also shed light on the mechanisms controlling the position of the cytokinetic furrow. Original studies in sand dollar eggs showed that the position of the centrosomes is a key determinant of the cytokinetic furrow position (*Rappaport, 1961*); later studies in *C. elegans* found that a second signal emanating from the spindle midzone also contributes to the positioning of the cytokinetic furrow (*Dechant and Glotzer, 2003*; *Bringmann and Hyman, 2005*). A role for chromosomes was, however, discarded in these two organisms, since midzone formation and cytokinesis did not require them. In contrast, in human cells, chromosomes stabilize microtubules of the midzone and thus favor the formation of a cytokinetic furrow (*Canman et al., 2003*). Here, we show that 2:1 cells only misplace the cytokinetic furrow in the presence of an asymmetric plate position in metaphase, implying that the position of the metaphase plate plays a crucial fine-tuning role in the positioning of the cytokinetic furrow. Future studies will have to test whether the metaphase plate acts via the microtubules of the midzone, or as recently postulated, by influencing the cortical populations of Anillin and Myosin in anaphase in a Ran-GTP-dependent manner (*Kiyomitsu and Cheeseman, 2013*).

The depletion of KIF2a/MCAK satisfies the SAC in 2:1 cells, but not in cells with unattached or tension-free kinetochore-microtubule attachments, indicating that the SAC responds to kinetochore–microtubule attachments defects less severe than lack of attachment or a tension defect. What might be these defects? Kinetochores in 2:1 cells bind a sufficient number of microtubules to form amphitelic attachments and stretch the two sister-kinetochores apart, but a number of kinetochores do not bind the full complement of stable microtubules required for SKAP loading. It is established that the SAC responds to detached kinetochores and is satisfied when kinetochores have bound the full set of microtubules. Based on our results, we postulate that the SAC also responds if a kinetochore is only bound by a fraction of the full set of microtubules. Such a lack of full occupancy would delay anaphase onset, such as in 2:1 cells; it would also explain the tendency for a higher rate of chromosome segregation errors in 2:1 cells treated with an Mps1 inhibitor or KIF2a/MCAK siRNAs. Stabilizing those kinetochore–microtubules by inhibiting Aurora-B or depleting KIF2a/MCAK

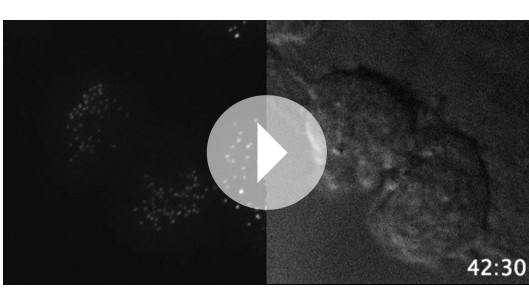

**Video 8.** Laser-ablated 2:1 HeLa cell expressing eGFP-centrin1 (centriole marker) and eGFP-CENPA (kinetochore marker) treated in metaphase with an Mps1 inhibitor. Shown is the GFP-fluorescence channel (left) and the DIC channel (right). Note how the cell divides in an asymmetric manner.

establishes full binding of microtubules, SKAP loading, and satisfies the SAC, allowing cells to enter anaphase. This suggests a SAC that is more sensitive than a checkpoint that only senses detached kinetochores or kinetochores that become detached due to a tension defect. A SAC that detects such minor defects in kinetochore–microtubule occupancy caused by an imbalance of microtubule stability within the spindle would be able to indirectly probe for plate positioning, giving cells time to correct this imbalance and ensure a symmetric metaphase plate position. Such graded response to microtubule occupancy within a kinetochore complements studies showing that the SAC acts in a graded manner when it comes to the number of unattached kinetochores (*Collin et al., 2013*; *Dick and Gerlich, 2013*).

## Materials and methods

### Cell culture, siRNA, and drug treatments

HeLa cells were grown in Dulbecco's modified medium containing 10% Fetal Calf Serum (FCS), 100 U/ml penicillin, 100 mg/ml streptomycin, at 37°C with 5% $CO_2$ in a humidified incubator. HeLa eGFP-centrin1/eGFP-CENPA, eGFP-Centrin1/mRFP-α-tubulin, and eGFP-Centrin2/H2B-mRFP (kind gift of U Kutay, ETH) cells were further maintained in 250 ng/ml puromycin and 250 μg/ml G418. Live-cell imaging experiments were performed at 37°C in Lab-Tek II chambers (Thermo Fischer, Switzerland) with Leibovitz L-15 medium containing 10% FCS. SiRNA oligonucleotides (Invitrogen and Thermo Fisher, Switzerland) against control (Scrambled), Sas-6, Mad2, Kid1, KIF2a, and MCAK were transfected using Oligofectamine (Invitrogen; *Meraldi et al., 2004*; *Ganem et al., 2005*; *Leidel et al., 2005*; *Wandke et al., 2012*; *Mchedlishvili et al., 2012*). Mad2 and Kid1 depletion had been previously validated in our laboratory (*Meraldi et al., 2004*; *Wandke et al., 2012*), Sas-6 depletion was validated by counting centrioles in all experiments; Mad2 and Sas-6 co-depletion was additionally validated by immunoblotting (*Figure 2—figure supplement 1*); KIF2a/MCAK depletion in Sas-6-depleted cells was validated by immunoblotting (*Figure 2—figure supplement 1*). To exclude off-target effects, KIF2a and MCAK were also depleted with an alternative set of pooled siRNAs (ON-TARGETplus Human KIF2C (11,004) and ON-TARGETplus Human KIF2a (3796) siRNA—SMARTpools; GE Healthcare, Switzerland). For drug treatments, cells were treated with 100 μM monastrol for 3 hr, 1 μM MG132 for 1 hr, or 10 nM taxol (all Sigma-Aldrich, Switzerland) for 15 min before fixation for immunofluorescence or live-cell imaging. The Mps1 inhibitors MPS1-IN-1 (10 μM; kind gift of NS Gray; [*Kwiatkowski et al., 2010*]) or Reversine (10 μM, Sigma–Aldrich), or the Aurora-B inhibitor ZM1 (2 μM, Tocris, United Kingdom) were added to metaphase cells during live-cell imaging. To determine the response to spindle poisons, cells were treated for 16 hr with combinations of 100 ng/ml nocodazole, 100 μM monastrol, 10 μM MPS1-IN-1 or 2 μM ZM1, and the percentage of mitotic cells determined by phase contrast microscopy.

### Immunofluorescence

Cells were fixed with methanol at −20°C for 6 min, or with 20 mM PIPES(pH 6.8), 10 mM EGTA, 1 mM $MgCl_2$, 0.2% Triton X-100, 4% formaldehyde for 7 min at room temperature. For the cold-stable assay, cells were incubated in cold medium whilst placed on ice for 7 min. The following primary antibodies were used: rabbit anti-Mad2 (1:1000; Bethyl); mouse anti-α-tubulin (1:10,000) and rabbit anti-γ-tubulin (1:2000; both Sigma–Aldrich); mouse anti-HEC1 (1:1000), mouse anti-pericentrin (1:2000, kind gift of U Kutay), rabbit anti-ninein (1:500), mouse anti-TPX-2 (1:250), and mouse anti-centrobin (1:1000; all Abcam, United Kingdom); mouse anti-p150^Glued (1:500; Becton Dickinson, Switzerland); rabbit anti-phospho-Aurora-A (1:1000; Cell signalling, Danvers, MA); rabbit anti-centrin (1:1000) and affinity-purified rabbit anti-SKAP (1 mg/ml; [*Schmidt et al., 2010*]; both gifts of I Cheeseman). Cross-adsorbed secondary antibodies were used (Invitrogen). Three-dimensional image stacks of mitotic cells were acquired in 0.2-μm steps using a 100× NA 1.4 objective on an Olympus DeltaVision microscope (GE Healthcare) equipped with a DAPI/FITC/TRITC/CY5 filter set (Chroma, Bellow Falls, VT) and a CoolSNAP HQ camera (Roper Scientific, Tucson, AZ). For quantitative measurements, 3D image stacks were deconvolved with SoftWorx (GE Healthcare) and quantified with SoftWorx, Imaris (Bitplane, Switzerland) or ImageJ. Images were mounted as figures using Adobe Illustrator. Kinetochore protein intensities were measured as a ratio to the CREST signal as described (*McClelland et al., 2007*). To monitor the polar ejection force, the distance between centrosomes and kinetochores was measured as described (*Wandke et al., 2012*).

### Live-cell imaging

For mitotic timing experiments, cells were recorded every 3 or 4 min as three-dimensional image stacks (12 × 1 μm steps using a 60× 1.4 NA objective, or 7 × 2 μm stacks using a 40× 1.3 NA objective) on an Olympus DeltaVision microscope equipped with a GFP/mRFP filter set (Chroma) and a CoolSNAP HQ camera. To monitor cell contours, cells were illuminated with white light and recorded by phase-contrast microscopy. Time-lapse videos were visualized in Softworx to quantify mitotic timing and to detect rotating spindles.

## Kinetochore tracking, metaphase plate position, spindle positioning, intra-kinetochore distance, and SKAP assays

For kinetochore tracking, plate width and plate position experiments, fluorescence time-lapse imaging of metaphase HeLa eGFP–centrin1/eGFP-CENPA cells was recorded with a 100× 1.4 NA objective on an Olympus DeltaVision microscope. 35 Z-sections 0.5 μm apart were acquired with a sampling rate of 7.5 s for a total duration of 5 min. Three-dimensional image stacks were deconvolved with SoftWorx and subjected to the kinetochore tracking assay analysis run in MATLAB (The Math Works, Inc, Natick, MA), to asses inter-kinetochore distances and kinetochore oscillations (*Jaqaman et al., 2010*). The tracking assay was also used to quantify the length of the two half-spindles: the tracking assay estimates the metaphase plate by fitting a plane to the calculated kinetochore positions; metaphase plate position relative to the spindle poles was calculated using a custom MATLAB function that detects centrioles and calculates plate position as the intersection of the fitted plane with the spindle axis. The earliest time point data of each cell imaged was used for plate position and inter-kinetochore distance analysis to ensure that data come from early metaphase cells. To measure plate position at anaphase and to better visualize the centering mechanisms, we used a temporal resolution of 30 s and applied our combined kinetochore and centrosome tracking analysis. Videos were manually screened for the presence of chromosome segregation errors. To determine spindle positions within cells, we used the centrosome positions to determine the center of the spindle (equidistant to both centrosomes) and compared it to the cell center, which was determined using phase contrast images (point on the spindle axis that is equidistant to both cell cortexes). To measure intra-kinetochore distances and SKAP signals, HeLa eGFP–centrin1/eGFP-CENPA cells were fixed and stained with anti-HEC1 and anti-SKAP antibodies, respectively. Three-dimensional image stacks of fixed cells were subjected to the kinetochore tracking assay for sister–kinetochore pair identification. Imaris was used in conjunction with a custom MATLAB function (*Source Code 1*) to measure the HEC1 and SKAP signals of the detected sister-kinetochores.

## Laser ablation experiments

Centriole ablations were carried out by 2–4 series (10 Hz repetition rate) of second-harmonic, single-mode, 532-nm pulses of an Nd:YAG laser (ULTRA-CFR TEM00 Nd:YAG from Big Sky Laser, Quantel, United Kingdom). The pulse width was 8 ns and the pulse energy used was 1.5–2 μJ. A more detailed description of the laser-microsurgery unit can be found in (*Pereira et al., 2009*). Imaging and laser focusing was performed using a 100× 1.4 NA plan-Apochromatic DIC objective on a Nikon TE2000U inverted microscope equipped with a Yokogawa CSU-X1 spinning-disk confocal head and an iXon[EM]+ Electron Multiplying CCD camera.

## Statistical methods

Statistical analyses were performed in R 2.15.0. Unpaired t-tests with Welch's correction and Mann–Whitney U tests (against 2:2 cells) were carried out to check for the statistical significance of normal and non-normal distributed data, respectively. Count data were analyzed using the Fisher's Exact test. Levene's test of equality of variance was used to check for equal variances, using the lawstat package ($\alpha = 0.05$). Graphs were plotted in R using the ggplot2 package and mounted in Adobe Illustrator.

## Acknowledgements

We thank the LMC of the ETH Zurich for microscopy support, U Kutay for the HeLa H2B-mRFP/eGFP-centrin2 cell line (ETHZ, CH), I Cheeseman for the SKAP and centrin antibodies (MIT, USA), Nathaniel Gray for the Mps1 inhibitor (Harvard, USA), and Ed Harry (Univ. of Warwick, UK) for adapting the tracking code to Nikon files. We thank J Pines (Univ. of Cambridge, UK), Andrew McAinsh (Univ. of Warwick, UK), Monica Gotta (Univ. of Geneva, CH), and the Meraldi lab members for critical discussions. PM was funded by an SNF-project grant, the Swiss Cancer league, the ETH Zurich, the University of Geneva and the Louis-Jeantet Foundation, CHT was supported by a short-term EMBO fellowship and IG by a Böhringer Ingelheim fellowship. CHT and IG are part of the MLS PhD School. HM is funded by the seventh framework program grant PRECISE from the European Research Council.

## Additional information

### Funding

| Funder | Grant reference | Author |
|--------|-----------------|--------|
| Schweizerische Herzstiftung (Swiss Heart Foundation) | 3100A0-120728 | Patrick Meraldi |
| Eidgenössische Technische Hochschule Zürich (Federal Institute of Technology Zurich) | | Patrick Meraldi |
| Université de Genève | | Patrick Meraldi |
| Louis-Jeantet Foundation (Fondation Louis-Jeantet) | | Patrick Meraldi |
| EMBO | | Chia Huei Tan |
| Boehringer Ingelheim Fonds (BIF) | | Ivana Gasic |
| European Research Council (ERC) | | Helder Maiato |

The funders had no role in study design, data collection and interpretation, or the decision to submit the work for publication.

### Author contributions

CHT, Acquisition of data, Analysis and interpretation of data, Drafting or revising the article; IG, SPH-R, DD, MB, HM, Acquisition of data, Analysis and interpretation of data; PM, Conception and design, Acquisition of data, Analysis and interpretation of data, Drafting or revising the article

### Author ORCIDs

Chia Huei Tan, http://orcid.org/0000-0002-5561-6246

## Additional files

### Supplementary file

• Source code 1. Custom built software in Matlab.

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
