## [Decision Letter]

Thank you for sending your work entitled “The equatorial position of the metaphase plate ensures symmetric cell divisions” for consideration at *eLife*. Your article has been favorably evaluated by Tony Hunter (Senior editor), a Reviewing editor, and three reviewers.

The Reviewing editor and the other reviewers discussed their comments before we reached this decision, and the Reviewing editor has assembled the following comments to help you prepare a revised submission.

There is a general consensus that this paper is interesting and worthy of publication in *eLife* but a number of concerns must first be addressed. In particular, the authors should:

1) Show how active the centering mechanism is with more time-lapse images. The authors should also show an R vs. time graph for all their individual traces.

2) Provide stronger arguments that the asymmetry of the plate in 2:1 cells is due to differences in minus end stability.

3) Improve the evidence that SKAP loading is a read-out for stability. The authors should also show data to clarify that SKAP loading is specifically reduced on the 1-centriole side, instead of sometimes being reduced on the 1-centriole side, and sometimes on the 2-centriole side.

4) Provide more data in support of the reproducibility and robustness of the laser ablation assay, and explain the change in spindle length after centriole ablation.

5) Back-up their claim that an asymmetric plate causes an asymmetric division.

6) Validate their RNAi and rescue experiments.

In addition to these concerns the authors should comment on possible differences between laser ablation and Sas-6 depletion, and more clearly explain how they compute asymmetry. The authors should also make it explicit that the delay in mitosis observed in cells whose metaphase plate is off-centre is a consequence of an imbalance in microtubule forces resulting from kinetochore-microtubule occupancy that is monitored by the SAC.

---

## [Author Response]

*1) Show how active the centering mechanism is with more time-lapse images. The authors should also show an R vs. time graph for all their individual traces*.

The R-values presented in the original manuscript were calculated using an automated tracking assay for kinetochores (GFP-CENP-A) and centrosomes (GFP-centrin1) recorded with a 7.5 seconds resolution. We knew from previous studies that this temporal resolution was necessary to catch record and analyze chromosome movements in metaphase (Jaqaman et al., J. Cell Biol. 2010). To avoid photo-toxicity we were however forced to work with low intensities and to limit our movies to 5 minutes. This meant that only a subset (10-15%) of recorded cells entered anaphase within those 5 minutes, and that it is therefore not possible to visualize the full scale of the plate positioning correction, which often lasts more than 5 minutes, in such movies. Our representation of R is therefore built on a population analysis of cells either early in metaphase or just before anaphase. This population analysis shows that just before anaphase onset, cells have on average a metaphase plate that is located more precisely in the middle of the spindle.

To improve the manuscript we first explain this fact in the text (subsection “Cells center the metaphase plate position before anaphase onset”); second, to better document the metaphase plate centering process we also recorded 15 minutes movies of 2:1 cells using a temporal resolution of 30s. We show in the novel Figure 1 the R-values over time for 15 single cells, as well as the median R and the 95% confidence interval for the cell population as they approach anaphase. These experiments directly visualize and validate the existence of a centering mechanism, providing a clearer representation to the reader.

*2) Provide stronger arguments that the asymmetry of the plate in 2:1 cells is due to differences in minus end stability*.

First, we would like to point out that our data showed that depletion of KIF2a/MCAK only partially corrects the asymmetry of the metaphase plate position in 2:1 cells. This suggests that the difference in microtubule stability is an important factor in the asymmetry of the plate, but certainly not the only one. We have now made this fact clearer in our manuscript (subsection “An asymmetric distribution of centrioles leads to an imbalance of microtubule stability”).

Second, to further support our hypothesis of a differential minus-end stability that causes the asymmetric plate positioning in 2:1 cells, we measured with a cold-stable assay minus-end stability in 2:1 cells after a 1 hour MG132 treatment. Since MG132 treatment leads to a symmetric position of the metaphase plate, our hypothesis predicted an attenuated difference in minus end stability between the 1- and 2-centriole pole. This is exactly what we found, as now shown in the novel Figure 3.

To more directly test our hypothesis we also tested whether a brief treatment with taxol, a drug that stabilizes microtubules, would correct metaphase plate position, as seen after KIF2a/MCAK depletion. As shown in the novel Figure 3, taxol treatment largely corrects the metaphase plate position, validating our hypothesis that the asymmetric plate position is caused by differences in microtubule stability.

*3) Improve the evidence that SKAP loading is a read-out for stability. The authors should also show data to clarify that SKAP loading is specifically reduced on the 1-centriole side, instead of sometimes being reduced on the 1-centriole side, and sometimes on the 2-centriole side*.

Our interpretation of SKAP as a marker for microtubule stability is not our claim, but a conclusion of Schmidt et al., JCB 2010, which showed that SKAP levels are inversely proportional to Aurora-B levels at kinetochores, which is itself inversely proportional to microtubule stability. This model is supported by our experiments showing that KIF2a/MCAK depletion in 2:1 cells restores normal SKAP levels at kinetochore, despite the presence of a partially asymmetric plate position. Since KIF2a and MCAK are both microtubule depolymerases that destabilize kinetochore-microtubules we concluded that SKAP is a read-out for microtubule stability. To fully validate this hypothesis we now in addition briefly treated 2:1 cells with 10 nM taxol and stained for SKAP. We found that 100% of the kinetochores were SKAP-positive, confirming that it is a read-out for k-fiber stability (novel Figure 4).

With regard to the second question, we find that SKAP is not systemically absent on the 1-centriole-side or the 2-centriole-side, but that the frequency of the absence is the same for both sides. We do not think that this invalidates our model, as the SKAP read-out just indicates a reduced stability and higher dynamics of kinetochore-microtubules on both sides of the sister-kinetochores, consistent with the fact that these kinetochores are over time moving and correcting their position towards the middle of the spindle. This is now pointed out in our manuscript (subsection “2:1 cells form amphitelic kinetochore-microtubule attachments that take longer to stabilize”)

*4) Provide more data in support of the reproducibility and robustness of the laser ablation assay, and explain the change in spindle length after centriole ablation*.

To provide a better overview of our laser ablation experiment, we now show the individual traces of R over time (before and after ablation) for 11 cells, in which a single centriole was ablated, and we plot the median of this cell population (novel Figure 4). As a negative control we also plot the median R over time for control-ablated cells (laser pulse in the cytoplasm). Figure 4 shows how R increases after destruction of a single centriole, leading to an asymmetric position of the metaphase plate, consistent with our Sas-6 depletion experiments. Moreover, we now provide a novel Supplementary movie 4, visualizing how the ablation of a single centriole creates an asymmetric position of the metaphase plate.

Since such laser-ablation experiments are technically very challenging, and not easy to combine with a second assay, we used mainly Sas-6 depletion to investigate how the removal of a single centriole affects spindle length, showing how it leads to an imbalance of microtubule dynamics (see also point 2 for the additional experiments).

*5) Back-up their claim that an asymmetric plate causes an asymmetric division*.

Our claim that an asymmetric plate position at anaphase onset was supported by two independent experiments: both KIF2a/MCAK depletion and Mps1 inhibition allow 2:1 cells to enter anaphase with asymmetric spindles resulting in an asymmetric cell division. Importantly, when Mps1 was inhibited in late metaphase cells, when the plate has reached a symmetric plate position, we only observed symmetric cell division. Since these cells are, apart from the plate position, identical to the 2:1 cells that divide asymmetrically, we concluded that the plate position is a key determinant for the (a)symmetry of cell division.

To fully confirm this claim, we have now performed a second back-up experiment, in which we created 2:1 cells by laser-ablation (to create an asymmetric plate position), before forcing cells into anaphase by abrogating the spindle assembly checkpoint with an Mps1 inhibitor. We found that 4 out of 6 laser-ablated 2:1 cells underwent an asymmetric cell division (ratio between daughter cell size of more than 1.2, a ratio never seen in normal cells; this study but also [19]). In contrast 6 out of 6 control-ablated cells had a ratio smaller than 1.08. (p-value in unpaired t-test with Welch correction p = 0.0451; new Figure 6, and Supplementary movies 6 and 7). Despite the low numbers – laser-ablation experiments are technically very challenging – these experiments validate our initial claim.

*6) Validate their RNAi and rescue experiments*.

While our initial manuscript was based mainly on functional validations (e.g. number of centrioles for Sas-6 depletion), we have now validated in addition every key siRNA (Mad2, Sas-6, KIF2a and MCAK) by immunoblotting (Figure 2—figure supplement 1; Figure 5—figure supplement 1).

Moreover, the reviewers were worried about potential siRNA off-target effects, reason for which they asked for siRNA rescue experiments. Since such rescue experiments are technically extremely challenging when analyzing double or triple siRNAs and very difficult to analyze in a conclusive manner, we have instead performed a number of complementary experiments for each siRNA or provide additional explanations.

Sas-6 siRNA: one possibility is that Sas-6 siRNA could affect the position of the metaphase plate, microtubule stability and SKAP loading on kinetochores and/or lead to a spindle checkpoint dependent anaphase delay as an off-target effect. However, this is not possible, since such an off-target effect would also be present in 1:1 cells that have been subjected to the same siRNA. 1:1 cells show, however, none of the phenotypes seen in 2:1 cells, thus excluding any off target effect. Moreover, we confirm the effects of Sas-6 depletion on spindle positioning and asymmetric cell division using laser-ablation, as an alternative tool to create 2:1 cells (e.g. new Figure 6).

Mad2 siRNA: this siRNA is used as a known tool to inactivate the spindle checkpoint (Figure 2). Instead of performing a rescue experiment, which is very difficult since mitotic timing is sensitive to minor changes in Mad2 levels, we repeated the Sas-6 depletion in the presence of an Mps1 inhibitor to inhibit the spindle assembly checkpoint in an independent manner. These experiments validated our Mad2 depletion experiments (Figure 2—figure supplement 2).

Kid siRNA: this siRNA was used as a positive control for a disruption of the polar ejection force, based on previous literature. It has been previously validated in our and other laboratories (Wandke et al., JCB 2012), and we functionally validate it as a positive control in Figure 3. In contrast, we find that Sas-6 depletion has no effect on the polar ejection force. Therefore, we cannot test for an off-target effect.

KIF2a/MCAK depletion: this double depletion has been validated in multiple publications (in particular from the Duane Compton laboratory), and the phenotype we report is consistent with the published literature. The novel aspect of our manuscript comes from co-depletion of Sas-6 siRNA in 2:1 cells. While we appreciate that a siRNA rescue experiment would be informative, we believe that a triple siRNA rescue experiment is nearly impossible, particularly since the overexpression of either of these microtubule depolymerases is toxic. To nevertheless exclude potential off-target effects, we repeated our experiments with an alternative set of KIF2a and MCAK siRNAs. As shown in Figure 5—figure supplement 1, this alternative set of KIF2a/MCAK siRNAs rescued the mitotic delay seen in 2:1 cells, confirming our experiments with the original KIF2a/MCAK siRNAs. Moreover, we also validated in the same Figure the new siRNA treatment by immunoblotting. Since this experiment indicated that both the old and the new siRNA against KIF2a only led to a partial depletion of KIF2a (50%), we further tested whether depletion of MCAK alone would be sufficient to overcome the mitotic arrest seen in 2:1 cells (Figure 5—figure supplement 1). Since this is not the case, we conclude that KIF2a co-depletion is essential to overcome the mitotic delay.

Finally, we emphasize that other claims associated with KIF2A/MCAK co- depletion are fully backed up by independent experiments:

Our claim that stabilization of kinetochore microtubules in 2:1 cells overcomes the spindle assembly checkpoint delay and allows anaphase entry with an asymmetric plate position is confirmed by the Aurora-B inhibition experiment;

Our claim that 2:1/KIF2A/MCAK-depleted cells that enter anaphase with asymmetrically positioned metaphase plate undergo an asymmetric cell division is validated by our Mps1 inhibition experiments performed on Sas-6 depleted and on laser-ablated 2:1 cells.

*In addition to these concerns the authors should comment on possible differences between laser ablation and Sas-6 depletion, and more clearly explain how they compute asymmetry. The authors should also make it explicit that the delay in mitosis observed in cells whose metaphase plate is off-centre is a consequence of an imbalance in microtubule forces resulting from kinetochore-microtubule occupancy that is monitored by the SAC*.

We envisage two potential reasons for the stronger phenotype seen in 2:1 cells obtained through laser ablation as compared to Sas-6 depletion. First, since Sas-6 depleted cells lose a centriole in interphase, they might be better able to partially compensate for this loss as they gradually build up the mitotic spindle. Second, it is likely that the laser pulse on a mitotic spindle pole not only impairs the centriole, but that it also disrupts biochemical activities associated with the mitotic spindle pole, such as microtubule-depolymerases, which are able to exert a pulling force on k-fibers (Meunier et al., Nat. Cell Biol, 2011). Both potential explanations are now mentioned in the text (subsection “2:1 cells form amphitelic kinetochore-microtubule attachments that take longer to stabilize”).

We now also explain in the text more explicitly, how we calculate R (subsection “Cells center the metaphase plate position before anaphase onset”).

Finally, we now more explicitly link the occupancy of kinetochores with the activation of the spindle assembly checkpoint in 2:1 cells (see Discussion, fourth paragraph).